# Understanding soil selenium accumulation and bioavailability through size resolved and elemental characterization of soil extracts

Julie Tolu [1,2] ✉, Sylvain Bouchet[1,2], Julian Helfenstein[3,5], Olivia Hausheer[1,2], Sarah Chékifi[1,2], Emmanuel Frossard[3], Federica Tamburini[3], Oliver A. Chadwick[4] & Lenny H. E. Winkel [1,2] ✉

Dietary deficiency of selenium is a global health threat related to low selenium concentrations in crops. Despite the chemical similarity of selenium to the two more abundantly studied elements sulfur and arsenic, the understanding of its accumulation in soils and availability for plants is limited. The lack of understanding of soil selenium cycling is largely due to the unavailability of methods to characterize selenium species in soils, especially the organic ones. Here we develop a size-resolved multi-elemental method using liquid chromatography and elemental mass spectrometry, which enables an advanced characterization of selenium, sulfur, and arsenic species in soil extracts. We apply the analytical approach to soils sampled along the Kohala rainfall gradient on Big Island (Hawaii), which cover a large range of organic carbon and (oxy)hydroxides contents. Similarly to sulfur but contrarily to arsenic, a large fraction of selenium is found associated with organic matter in these soils. However, while sulfur and arsenic are predominantly found as oxyanions in water extracts, selenium mainly exists as small hydrophilic organic compounds. Combining Kohala soil speciation data with concentrations in parent rock and plants further suggests that selenium association with organic matter limits its mobility in soils and availability for plants.

Human deficiencies of micronutrients, including vitamins and trace elements, often referred to as "hidden hunger", is a critical obstacle to the United Nations' second Sustainable Development Goal to 'achieve food security and improved nutrition' by 2030[1]. These deficiencies potentially affect already more than 2 billion people worldwide[2] and are projected to increase due to soil ecosystems deterioration and reduction of anthropogenic atmospheric inputs[3–5]. Selenium (Se) is such an essential micronutrient for human and animal health[6]. Up to 1

billion people worldwide are estimated to have inadequate Se intakes, largely due to low concentrations in staple food crops[7,8]. Scant Se levels in food-crops result from generally low soil Se concentrations combined with a limited availability for plants, for which Se is not essential.

The retention and plant availability of micronutrients in soils are controlled, although not exclusively, by their speciation. Various inorganic Se species exist in soils, i.e., selenate ($SeO_4^{2-}$; oxidation state,

[1]Eawag, Swiss Federal Institute of Aquatic Science and Technology, Department of Water Resources and Drinking Water (W+T), Überlandstrasse 133, 8600 Dübendorf, Switzerland. [2]ETH Zurich, Swiss Federal Institute of Technology, Department of Environment Systems Sciences (D-USYS), Institute of Bio-geochemistry and Pollutant Dynamics (IBP), Group of Inorganic Environmental Geochemistry, Universitätstrasse 16, 8092 Zurich, Switzerland. [3]ETH Zurich, Swiss Federal Institute of Technology, Department of Environment Systems Sciences (D-USYS), Institute of Agricultural Sciences (IAS), Group of Plant Nutrition, Eschikon 33, 8315 Lindau, Switzerland. [4]Department of Geography, University of California, Santa Barbara, CA 93106, USA. [5]Present address: Soil Geography and Landscape Group, Wageningen University, 6700 AA Wageningen, The Netherlands. ✉e-mail: julie.tolu@eawag.ch; lenny.winkel@eawag.ch

VI) and selenite ($HSeO_3^-$/$SeO_3^{2-}$; IV) present as free oxyanions in soil solution or sorbed onto mineral (nano)particles, Se(0) (nano)particles, and metal selenides (-II)[9,10]. For simplicity, we denote thereafter selenite as $SeO_3^{2-}$(IV). Free and mineral-adsorbed Se oxyanions are considered mobile and plant-available whereas Se(0) (nano)particles, metal selenides and Se oxyanions co-precipitated with minerals are much less, or not at all, mobile and plant available[9,11]. Organic Se species are also present in soils[12–15], but they are only indirectly characterized (by "selective" extractions or difference between total and inorganic Se) and their behavior remain unclear in different aspects[16]. While some studies suggested an immobilization and reduced plant availability of organic Se[8,17–19], others proposed it as an important plant-available Se pool through its direct uptake (e.g., seleno-amino acids), its oxidation into plant-available Se oxyanions, and/or the release of complexed Se oxyanions into solution during organic matter degradation[20–23]. Due to a lack of methods dedicated to organic Se, the pathways of soil organic Se formation and degradation remain unrevealed. Nevertheless, hypotheses about the nature of organic Se were made based on its chemical similarity to sulfur (S) and arsenic (As)[11,12,16,24]. Similarly to S[25], organic Se may consist of Se proteins and metabolites and/or Se covalently bound to soil organic matter (SOM) after incorporation of microbially produced hydrogen selenide(-II) into SOM. Similarly to As[26,27], organic Se may consist of oxyanions bound to SOM (through ternary complexation) and/or adsorbed onto mineral (nano)particles coated by SOM. However, the combined speciation of Se, S and As in soils has not been investigated previously.

Determining soil Se speciation is challenging because of the generally low Se concentrations. The global average soil Se concentration (0.32 mg kg$^{-1}$;[4]) is well below the detection limit of solid-phase techniques such as X-ray absorption spectroscopy (XAS), which can only be applied to Se-contaminated (seleniferous) soils[15,28,29]. For common soils, various selective extractions protocols are used, generally targeting water-soluble, mineral-adsorbed Se oxyanions, and organic Se (Supplementary Note 1). However, selectivity of these extractions is questionable[28–30] (Supplementary Note 2). Furthermore, speciation analysis of soil extracts using selective reduction of Se(IV)[19,20,22,23] or chromatographic separation of Se oxyanions and amino acids was sometimes performed[13,14,31,32], but the former only informs on Se oxidation states whereas a large share of extracted Se species (20–100%) remains unrecovered by commonly used liquid chromatographic methods. Currently, the predominant nature (organic vs mineral) and characteristics (e.g., size, charge) of Se species present in soil extracts remain largely unknown.

In this study, we first develop a method combining size exclusion chromatography (SEC) with on-line organic matter (ultraviolet detector; UV) and multi-elemental (inductively coupled plasma tandem mass spectrometry; ICP-MS/MS) detections to determine Se speciation in soil extracts. SEC-UV-ICP-MS has previously been used to probe metal associations with organic macromolecules and inorganic particles in model systems[24], compost leachates[33], freshwaters[34,35] and biological samples[36,37], and here we optimize it for 0.1 M NaOH and ultrapure water extracts of natural soils. NaOH is widely used to estimate organic Se quantity and characterize SOM, while ultrapure water is employed to assess mobile and plant-available Se[11,16,38] (Supplementary Notes 1 and 3). Our method achieves a high recovery of Se species in both types of extracts through quantifying small (organo)mineral nanoparticulate, organic and free anionic Se forms with semi-quantitative information on other elements. Secondly, we apply the developed method to a selection of contrasting soils and related Se, As and S speciation to their accumulation in soils and concentrations in plants. We have selected soils from the well-studied Kohala climate gradient (Hawaii) that developed on a ~150,000-year-old lava flow and under a steep rainfall gradient (~275–3123 mm y$^{-1}$)[39]. These volcanic soils are ideal to develop the SEC-UV-ICP-MS/MS method and to evaluate the extent and behavior of the organic versus mineral soil Se pools because they cover

a wide range in soil organic carbon (SOC) concentration while being rich in amorphous (oxy)hydroxide minerals[40–42]. We demonstrate a large association of Se to SOM, with variable proportions of larger aromatic versus smaller hydrophilic OM fractions, and a limited importance of mineral nanoparticles in these soils. In NaOH extracts, the speciation of Se clearly differs from As but is relatively similar to S speciation. In water extracts Se and S are however markedly different with S being mainly present as sulfate while Se predominantly exists as small hydrophilic organic Se (other than Se-amino acids). By combining soil, parent rock and plant data from Kohala, we infer that the dominant organic nature of Se in soils is pivotal to understand its accumulation in soils and low plant availability.

## Results and discussion

### Chemical properties and concentrations of Se, As, and S in Kohala soils

We selected 25 soil samples from different depths at six sites along the Kohala rainfall gradient (S1–S6). The soil samples cover A- and B horizons, with the horizon depths being different at each site (Supplementary Table 1). At the arid sites S1–S2 (rainfall, 275–316 mm y$^{-1}$)[39], there is only sparse vegetation and topsoils (0–10 cm) are eroded by wind[40,41,43]. Sub-humid sites S3–S4 (1340–1578 mm y$^{-1}$)[39], where mean annual precipitation is close to potential evapotranspiration, are subjected to intense chemical weathering[40,41,43], and S3 topsoil also experiences a strong enrichment in nutrients (e.g, phosphorus) through nutrient uplifting[44]. At humid sites S5–S6 (2163–3123 mm y$^{-1}$)[39], mean annual precipitation is well above potential evapotranspiration resulting, at site S6, in intensive leaching and periods of anoxic conditions during heavy rainfalls where rainfall rates outpace leaching rates leading to water saturation[40,41,43].

With respect to soil chemistry, the 25 selected samples cover a wide range in SOC concentrations (1–26 %), pH (4.2–7.1), and concentrations of crystalline and amorphous iron (Fe) (oxy)hydroxides (0.9–7.9% and 0.2–8.0%, respectively; Fig. 1a–c and Supplementary Table 2). As previously reported[40,41], SOC concentrations and proportions of secondary amorphous minerals increase along the rainfall gradient whereas proportions of primary crystalline minerals and pH decrease. Total soil Se concentrations (0.25–3.6 mg kg$^{-1}$; Fig. 1d), determined using microwave-assisted digestion (with a mixture of nitric acid, hydrofluoric acid, and hydrogen peroxide), are representative although in the upper range of common soil Se levels[4], for which analyses by solid-phase speciation techniques is hardly applicable. As and S concentrations range between 3 and 17 mg kg$^{-1}$ and 0.25 and 2.8 g kg$^{-1}$, respectively (Fig. 1e, f). Along the gradient, the concentrations of Se, As and S in the horizon A increase until the humid site S5, following the trends of SOC and amorphous Fe (oxy)hydroxides concentrations. Between S5 and S6 (the wettest site), the concentrations of As in horizon A clearly decrease showing a similar trend as amorphous Fe (oxy)hydroxides, whereas S concentrations in horizon A increase similarly to SOC concentrations. Like As, Se concentrations in horizon A decrease between S5 and S6 but to a lesser extent. In the horizon B, Se and S show similar trends than SOC while As is again similar to amorphous Fe (oxy)hydroxides.

### Characterization of Se species in soil extracts with SEC-UV-ICP-MS/MS

To obtain a size-resolved and multi-elemental characterization of Se species in soil extracts (filtered at 0.45 μm), we developed a SEC-UV-ICP-MS/MS method that includes a quantification of Se by on-line isotope dilution and peak deconvolution. We focused on NaOH extracts, targeting organic and mineral-adsorbed Se, and on ultrapure water extracts that are commonly used to estimate mobile and plant-available Se[11,16,29,30]. NaOH and water extracts of Kohala soils contain respectively, 2–80 and 0.8–12 μg(Se) L$^{-1}$, which account for 32-89 and

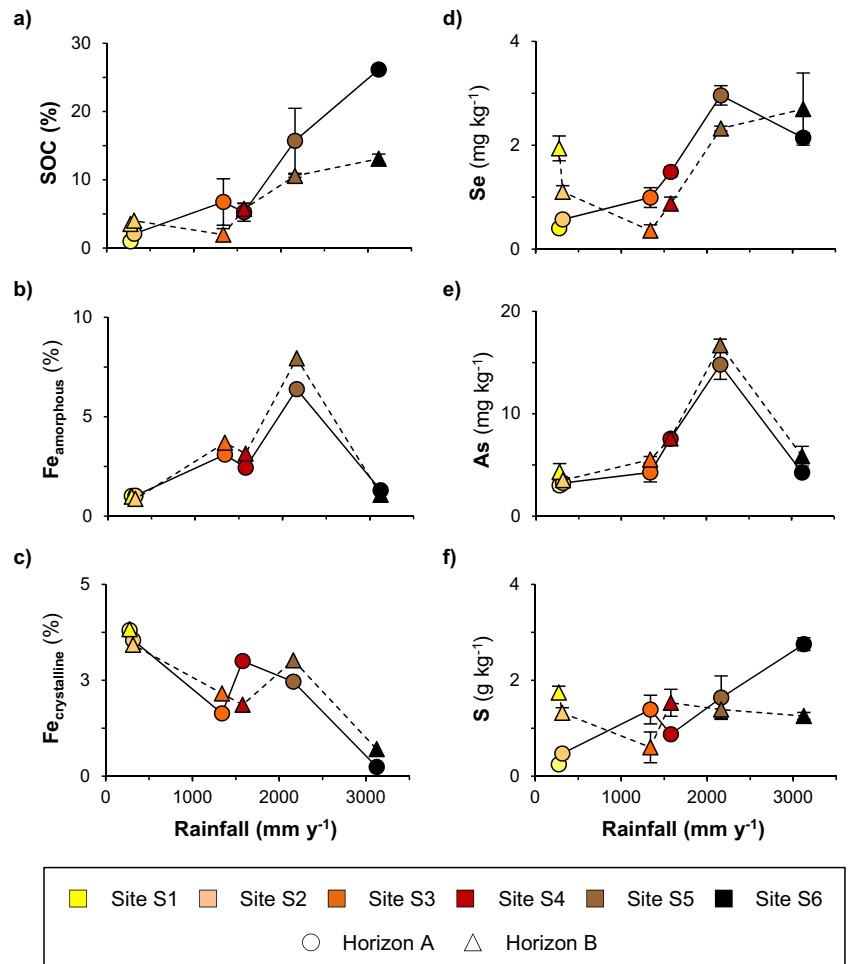

**Fig. 1 | Concentrations of soil organic carbon (SOC), amorphous and crystalline iron minerals ($Fe_{amorphous}$ and $Fe_{crystalline}$), selenium (Se), arsenic (As) and sulfur (S) along the Kohala rainfall gradient.** Correspondence between soil horizons and soil depths is given in Supplementary Table 1. The concentrations of $Fe_{amorphous}$ and $Fe_{crystalline}$ (**b** and **c**) were measured by Helfenstein et al.[67] on sampled soil horizon A and B without triplicate and thus no standard deviation value is available. The concentrations of SOC (**a**) and of total Se, As, and S (**d**–**f**) were determined in triplicate for each analyzed soil depth samples, and the presented standard deviations (error bars) consider both the analytical standard deviation (resulting from the analysis in triplicate) and the standard deviation resulting from averaging the measured concentrations for soil depth samples to obtain the concentrations for soil A and B horizons. Invisible error bars for SOC, Se, As, and S concentrations mean that they are within the symbol.

0.2–2.7% of total soil Se (Table 1), in line with reported extraction efficiencies (Supplementary Note 3). We tested different types of SEC columns and mobile phase (Supplementary Discussion 1, Supplementary Table 3). The best chromatographic resolution and recoveries of Se were obtained with a series of Shodex OH-Pak 803-SB and 802.5-SB columns (size separation, 0.3–100 kDa; <~40 nm) and a mobile phase containing 5 mmol $L^{-1}$ ammonium nitrate at pH 7 and 9.5 for water and NaOH extracts, respectively (Supplementary Discussions 1 and 2, Supplementary Tables 3 and 4, Supplementary Figs. 1–4).

From the combination of multiple parameters (size, UV absorbance, and multi-element intensity) offered by the optimized SEC-UV-ICP-MS/MS method, we could classify Se and other elements into five size and chemical fractions for both types of extracts (Fig. 2, Supplementary Discussion 3, and Supplementary Figs. 5–8). These fractions are: F1. (organo)mineral nanoparticles likely including Fe (oxy)hydroxides nanoparticles that are mobilized from soils and also potentially formed during NaOH extraction[45,46]; F2a. larger, more hydrophobic and negatively charged, Fe-enriched, aromatic OM in NaOH extracts, and F2b. larger and aliphatic, non-Fe enriched, OM in water extracts; F3. smaller, less hydrophobic and negatively charged, aromatic OM; F4. small hydrophilic OM; and F5. free Se, As, and S oxyanions.

Considering all soils, the Se recovery obtained with SEC accounts for 58–109% of total Se in water extracts ($Se_{water}$) and 95–112% of total Se in NaOH extracts ($Se_{NaOH}$; Table 1). In contrast, 40–100% of $Se_{water}$ and 38–90% of $Se_{NaOH}$ are unrecovered with the conventional method of anion exchange chromatography (AEC) coupled to ICP-MS/MS, with which only Se oxyanions were detected (Table 1). Importantly, the sum of concentrations of Se oxyanions obtained by SEC-UV-ICP-MS/MS matches the one obtained by AEC-ICP-MS/MS (Table 1, Supplementary Discussion 4, and Supplementary Fig. 9). Apart from Se, high SEC recoveries (>80%) were obtained for S, Cu, and Zn, known to be largely incorporated or complexed to OM[47–49] (Table 2). Recoveries are, however, lower for Fe, As and Pb, which are well-known to form or adsorb to mineral (nano)particles[34,45,46], indicating that soil extracts contain such fractions that are too large to be eluted by our selected SEC columns, i.e., larger than columns pore size of ~40 nm. Elemental quantification of filtrates of water and NaOH extracts (filtered at 20 nm) further indicates a significant fraction of mineral (nano)particles >20 nm, with the proportions of unrecovered Se by SEC in Kohala topsoil extracts matching those of Se in the <20 nm filtrates of these extracts (i.e., respectively 66–106 versus 66–100% of $Se_{water}$ and 96–104 versus 96–100% of $Se_{NaOH}$; Tables 1–2). The fraction of $Se_{water}$ that was not recovered by our SEC method (<44%) is thus likely

**Table 1 | Selenium (Se) concentrations and proportions in Kohala soil water and NaOH extracts, and comparison between Se oxyanions proportion and Se species recovery obtained with anion exchange chromatography (AEC) and size exclusion chromatography (SEC) coupled to ICP-MS/MS**

| Sample (depth; horizon) | Water extracts | | | | | |
| --- | --- | --- | --- | --- | --- | --- |
| | Se concentration ($\mu$g L$^{-1}$) | Se proportion[b] (% soil Se) | Se$_{oxyanions}$ by AEC[c] (% Se$_{water}$) | Unrecovered Se by AEC[c] (% Se$_{water}$) | Se$_{oxyanions}$ by SEC[d] (% Se$_{water}$) | Recovered Se by SEC[d] (% Se$_{water}$) |
| S1 (0–10 cm; A) | 1.0 ± 0.1 | 1.2 ± 0.1 | 26 ± 2 | 74 ± 8 | 24 ± 2 | 68 ± 6 |
| S1 (10–20 cm; B) | 2.1 ± 0.2 | 0.51 ± 0.08 | 60 ± 5 | 40 ± 10 | 70 ± 6 | 83 ± 8 |
| S2 (0–10 cm; A) | 1.3 ± 0.1 | 1.2 ± 0.2 | 23 ± 2 | 77 ± 8 | 19 ± 1 | 66 ± 4 |
| S2 (10–20 cm; B) | 2.3 ± 0.2 | 1.0 ± 0.1 | 30 ± 3 | 70 ± 9 | 35 ± 4 | 88 ± 9 |
| S3 (0–10 cm; A) | 5.8 ± 0.4 | 2.7 ± 0.2 | 14 ± 1 | 86 ± 9 | 15 ± 1 | 106 ± 12 |
| S3 (10–20 cm; A) | 3.2 ± 0.1 | 1.96 ± 0.08 | 23 ± 1 | 77 ± 5 | 15 ± 1 | 109 ± 4 |
| S3 (20–30 cm; B) | 1.9 ± 0.1 | 2.07 ± 0.08 | 19 ± 1 | 81 ± 4 | 13 ± 1 | 82 ± 3 |
| S3 (30–40 cm; B) | 1.4 ± 0.1 | 2.0 ± 0.5 | 35 ± 3 | 65 ± 9 | 35 ± 3 | 69 ± 5 |
| S3 (40–50 cm; B) | 0.8 ± 0.1 | 1.7 ± 0.5 | 33 ± 3 | 67 ± 12 | 34 ± 3 | 58 ± 5 |
| S4 (0–10 cm; A) | 3.8 ± 0.1 | 1.22 ± 0.04 | 3.9 ± 0.3 | 96 ± 5 | 12 ± 3 | 93 ± 6 |
| S4 (10–20 cm; A) | 3.0 ± 0.3 | 1.0 ± 0.1 | 5.5 ± 0.6 | 95 ± 14 | 13 ± 1 | 77 ± 8 |
| S4 (20–30 cm; A) | 2.5 ± 0.1 | 0.91 ± 0.06 | 6.7 ± 0.6 | 93 ± 6 | 8.5 ± 0.4 | 68 ± 3 |
| S4 (30–40 cm; B) | 1.6 ± 0.1 | 0.9 ± 0.1 | 10 ± 1 | 90 ± 11 | 14 ± 1 | 72 ± 6 |
| S4 (40–50 cm; B) | 1.59 ± 0.04 | 0.81 ± 0.04 | 14 ± 1 | 86 ± 4 | 27 ± 1 | 73 ± 2 |
| S4 (50–60 cm; B) | 0.83 ± 0.04 | 0.45 ± 0.03 | 17 ± 2 | 83 ± 7 | 42 ± 2 | 80 ± 4 |
| S4 (60–70 cm; B) | 0.9 ± 0.1 | 0.65 ± 0.08 | 25 ± 2 | 75 ± 8 | 35 ± 2 | 79 ± 5 |
| S5 (0–10 cm; A) | 8.3 ± 0.1 | 1.5 ± 0.2 | <0.5 | >99 | Undetected | 95 ± 3 |
| S5 (10–20 cm; A) | 6.5 ± 0.3 | 1.08 ± 0.07 | <3 | >99 | Undetected | 97 ± 4 |
| S5 (20–30 cm; B) | 2.34 ± 0.05 | 0.49 ± 0.04 | <6 | >97 | Undetected | 80 ± 2 |
| S5 (30–40 cm; B) | 2.54 ± 0.02 | 0.57 ± 0.04 | <7 | >97 | Undetected | 85 ± 1 |
| S6 (10–20 cm; A) | 12.1 ± 0.2 | 2.9 ± 0.1 | <7 | >99 | Undetected | 94 ± 3 |
| S6 (20–30 cm; B) | 1.9 ± 0.2 | 0.27 ± 0.02 | <0.7 | >97 | Undetected | 70 ± 6 |
| S6 (30–40 cm; B) | 0.9 ± 0.1 | 0.17 ± 0.03 | <0.9 | >94 | Undetected | 72 ± 7 |
| S6 (40–50 cm; B) | 0.9 ± 0.1 | 0.20 ± 0.03 | <3 | >93 | Undetected | 75 ± 7 |
| S6 (50–60 cm; B) | 0.8 ± 0.1 | 0.20 ± 0.02 | <2 | >93 | Undetected | 60 ± 6 |

| Sample (depth; horizon) | NaOH extracts | | | | | |
| --- | --- | --- | --- | --- | --- | --- |
| | Se concentration ($\mu$g L$^{-1}$) | Se proportion[b] (% soil Se) | Se$_{oxyanions}$ by AEC[c] (%Se$_{NaOH}$) | Unrecovered Se by AEC[c] (%Se$_{NaOH}$) | Se$_{oxyanions}$ by SEC[d] (%Se$_{NaOH}$) | Recovered Se by SEC[d] (%Se$_{NaOH}$) |
| S1 (0–10 cm; A) | 3.8 ± 0.4 | 32 ± 4 | 42 ± 4 | 58 ± 6 | 40 ± 2 | 102 ± 9 |
| S1 (10–20 cm; B) | 31.2 ± 0.6 | 53 ± 7 | 63 ± 5 | 37 ± 3 | 56.1 ± 0.7 | 108 ± 3 |
| S2 (0–10 cm; A) | 6.7 ± 0.2 | 39 ± 5 | 36.1 ± 0.9 | 64 ± 2 | 34.1 ± 0.9 | 102 ± 6 |
| S2 (10–20 cm; B) | 23 ± 1 | 66 ± 7 | 52 ± 2 | 48 ± 2 | 44.2 ± 0.5 | 102 ± 3 |
| S3 (0–10 cm; A) | 20 ± 1 | 59 ± 3 | 23 ± 1 | 77 ± 4 | 30.8 ± 0.4 | 101 ± 4 |
| S3 (10–20 cm; A) | 15.7 ± 0.2 | 61 ± 3 | 36 ± 2 | 64 ± 3 | 35.1 ± 0.6 | 97 ± 2 |
| S3 (20–30 cm; B) | 9 ± 1 | 67 ± 4 | 37 ± 2 | 63 ± 4 | 39.4 ± 0.8 | 95 ± 6 |
| S3 (30–40 cm; B) | 6.2 ± 0.6 | 57 ± 15 | 42 ± 12 | 58 ± 17 | 38.0 ± 0.8 | 95 ± 5 |
| S3 (40–50 cm; B) | 4.3 ± 0.4 | 56 ± 17 | 47 ± 7 | 53 ± 7 | 51 ± 4 | 99 ± 6 |
| S4 (0–10 cm; A) | 27 ± 1 | 60 ± 1 | 18.4 ± 0.4 | 82 ± 2 | 25.1 ± 0.4 | 96 ± 5 |
| S4 (10–20 cm; A) | 29.7 ± 0.6 | 66 ± 3 | 29 ± 1 | 71 ± 3 | 32.8 ± 0.5 | 100 ± 1 |
| S4 (20–30 cm; A) | 28.4 ± 0.4 | 70 ± 4 | 34 ± 1 | 66 ± 2 | 31.3 ± 0.4 | 103 ± 5 |
| S4 (30–40 cm; B) | 19 ± 1 | 68 ± 9 | 19 ± 1 | 81 ± 5 | 19.3 ± 0.3 | 101 ± 5 |
| S4 (40–50 cm; B) | 21.6 ± 0.6 | 69 ± 7 | 22 ± 2 | 78 ± 7 | 20.1 ± 0.3 | 105 ± 3 |
| S4 (50–60 cm; B) | 17.0 ± 0.8 | 62 ± 5 | 25 ± 2 | 75 ± 6 | 23.9 ± 0.4 | 108 ± 11 |
| S4 (60–70 cm; B) | 14.9 ± 0.6 | 69 ± 8 | 26 ± 1 | 74 ± 3 | 24.8 ± 0.4 | 112 ± 8 |
| S5 (0–10 cm; A) | 68 ± 3 | 76 ± 12 | 12 ± 1 | 88 ± 6 | 11.0 ± 0.2 | 99 ± 3 |
| S5 (10–20 cm; A) | 79 ± 7 | 86 ± 7 | 29 ± 3 | 71 ± 6 | 27.1 ± 0.4 | 97 ± 2 |
| S5 (20–30 cm; B) | 55 ± 4 | 72 ± 5 | 33 ± 2 | 67 ± 4 | 30.2 ± 0.6 | 99 ± 2 |
| S5 (30–40 cm; B) | 55 ± 5 | 79 ± 9 | 27 ± 2 | 73 ± 6 | 27.5 ± 0.5 | 96 ± 2 |
| S6 (10–20 cm; A) | 50 ± 4 | 77 ± 7 | 10 ± 2 | 90 ± 17 | 10.6 ± 0.2 | 102 ± 5 |
| S6 (20–30 cm; B) | 77 ± 3 | 70 ± 4 | 17 ± 1 | 83 ± 4 | 14.6 ± 0.2 | 101 ± 5 |
| S6 (30–40 cm; B) | 62 ± 1 | 70 ± 9 | 16 ± 1 | 84 ± 3 | 13.4 ± 0.2 | 99 ± 6 |

**Table 1 (continued) | Selenium (Se) concentrations and proportions in Kohala soil water and NaOH extracts, and comparison between Se oxyanions proportion and Se species recovery obtained with anion exchange chromatography (AEC) and size exclusion chromatography (SEC) coupled to ICP-MS/MS**

| Sample (depth; horizon) | NaOH extracts | | | | | |
| | Se concentra-tion (µg L$^{-1}$) | Se proportion$^b$ (% soil Se) | Se$_{oxyanions}$ by AEC$^c$ (%Se$_{NaOH}$) | Unrecovered Se by AEC$^c$ (%Se$_{NaOH}$) | Se$_{oxyanions}$ by SEC$^d$ (%Se$_{NaOH}$) | Recovered Se by SEC$^d$ (%Se$_{NaOH}$) |
|---|---|---|---|---|---|---|
| S6 (40–50 cm; B) | 53 ± 1 | 76 ± 12 | 15 ± 1 | 85 ± 6 | 11.5 ± 0.3 | 97 ± 5 |
| S6 (50–60 cm; B) | 45 ± 2 | 72 ± 4 | 15 ± 1 | 85 ± 5 | ± 0.2 | 97 ± 7 |

$^a$The standard deviations provided for the concentrations of Se in the extracts result from the ICP-MS/MS acquisition performed in triplicate.
$^b$The standard deviations provided for the proportions of Se in the extracts include the standard deviation obtained for the concentrations of Se in the extract (see $^a$) and the one obtained for soil Se concentrations (soils were digested in triplicate).
$^c$The standard deviations provided for the proportions of Se oxyanions and the Se recovery determined by AEC-ICP-MS/MS include the standard deviation resulting from the analysis by AEC-ICP-MS/MS of each extract in duplicate and the one obtained for the Se concentrations in the extracts (see $^a$).
$^d$The standard deviations provided for the proportions of Se oxyanions and the Se recovery determined by SEC-UV-ICP-MS/MS include the standard deviation resulting from the deconvolution of SEC Se peaks (using the peak analyzer function "Fit peak-pro" of Origin2018 software) and the standard deviation obtained for the Se concentrations in the extracts (see $^a$).

associated with >20–40 nm (nano)particles, and future studies are needed to better characterize this Se fraction in soil water extracts using a SEC column specific for (nano)particles[46] or field flow fractionation[45]. Additionally, although only Se(0) nanoparticles of 50–500 nm size have been found in contaminated soils or bacterial cultures[50–52], we cannot exclude that the fractions F2–4, which we assigned as organic Se, contain very small Se(0) nanoparticles. Altogether, our developed SEC-UV-ICP-MS/MS method enables a better understanding of Se cycling in soils and acquisition by plants as it allows for the identification and reproducible quantification (Supplementary Discussions 5 and 6, Supplementary Figs. 10 and 11) of organic Se of different size and chemical properties together with free Se oxyanions and Se associated with small (organo)mineral nanoparticles (<20–40 nm).

## Organic Se is a dominant form in Kohala soils

The distribution of Se among the various SEC fractions we identified in NaOH and water extracts of Kohala soils is presented for soil horizon A and B in Fig. 3. 11–56% of Se extracted by NaOH (Se$_{NaOH}$) exists as free Se oxyanions (F5; Fig. 3a), which were most likely adsorbed onto minerals[29,30]. Despite the presence of mineral, likely (oxy)hydroxides, nanoparticles in these extracts, Se associated with F1 only represents <3% of Se$_{NaOH}$. On the contrary, organic Se (sum of fractions F2–4) accounts for 44–89% of Se$_{NaOH}$. The proportion of alkaline-extractable organic Se thus ranges from 19% of soil Se in oxic, mineral soils (1% SOC) up to 69% in anoxic, organic-rich soils (26% SOC), and strongly correlates with SOC (Fig. 3b). Organic Se is also a dominant form in water extracts (13–95% of Se$_{water}$; Fig. 3c), where it significantly correlates with dissolved OC (range, 1.8–73 mg(C) L$^{-1}$; Fig. 3d). In contrast to NaOH extracts, in water extracts there is always some Se associated to (organo)mineral (nano)particles (fraction F1 plus Se fraction unrecovered by SEC; 23 ± 13% of Se$_{water}$), especially in soils from horizon B and from arid and sub-humid sites, but not always free Se oxyanions. Se oxyanions are indeed absent in soil horizons A and B at the humid, SOC-rich sites (S5–S6).

Among organic fractions, Se$_{NaOH}$ is mostly found in smaller, less hydrophobic and negatively charged, aromatic OM (F3, 44 ± 13% of Se$_{NaOH}$) as well as in small hydrophilic OM (F4, 19 ± 7%) and only to a minor extent in larger and negatively charged, Fe-enriched, aromatic OM (F2a, 9 ± 7%; Fig. 3a). Similarly, most of the organic Se$_{water}$ is present in small hydrophilic OM (44 ± 17% of Se$_{water}$), and in smaller, less hydrophobic and negatively charged aromatic OM (18 ± 12%; Fig. 3c). Only very little Se$_{water}$ is found in larger and aliphatic OM (F2b, 0.6 ± 1.2%). Potentially, the small hydrophilic organic Se fraction could include Se metabolites such as selenosugars (Se–C bound; only identified until now in Se-enriched plants[36]), but not SeMet, SeCys$_2$ and methane seleninic acid (MeSeOOH). Indeed, SeMet, SeCys$_2$, and MeSeOOH were not detected by AEC-ICP-MS/MS in Kohala soils

(Table 1) and the retention times of these Se species with SEC did not match with any peak of F4. Interestingly, the proportions of organic Se$_{NaOH}$ in F2a and F3 clearly increase from sites S1 to S6 whereas the proportion of small hydrophilic organic Se$_{NaOH}$ decreases between sites S1 and S6 from 38 to 7% (horizon A) and 25 to 11% (horizon B; Fig. 3a), and a similar trend is observed for carbon (Supplementary Fig. 12). At the driest sites (S1-2), the vegetation cover is scarce, which results in lower inputs of fresh OM than in (sub-)humid sites S3-6[40,41,43]. Additionally, previous lab-studies on natural OM observed that aerobic degradation of larger, more negatively charged and hydrophobic OM fractions produce small, more hydrophilic and/or non-aromatic compounds[53,54]. We therefore infer that the small hydrophilic organic Se fraction originates from Se association to small hydrophilic SOM produced by SOM degradation and/or from degradation of large, more hydrophobic and negatively charged organic Se.

Overall, our study precisely quantifies and characterizes alkaline-extractable organic Se, thus shedding insight on the degree and types of Se association to SOM in soils. Previous studies suggested that SOM could mainly play an indirect role on Se cycling through coating of Fe oxides (nano)particles to which Se oxyanions are adsorbed[14,31,55] and/or by favoring (a)biotic Se reduction into Se(0) nanoparticles[9,56,57], which commonly have sizes >50 nm[50–52]. Here, we demonstrate that over a large gradient of SOC content, variable mineral composition and redox conditions, organic Se is a dominant Se pool that consists of different size and chemical fractions.

## Extractability, size distribution and speciation largely differ between Se and As

Much less As is extracted from Kohala soils by NaOH than Se (26 ± 10% of soil As against 65 ± 12% of soil Se; Supplementary Fig. 13). Moreover, 26-54% of this As$_{NaOH}$ is found to be associated with SOM (Fig. 4a), accounting for 3–17% of soil As only (Fig. 4b) whereas 19–69% of soil Se is alkaline-extractable organic Se (Fig. 3b). Most of As$_{NaOH}$ is found associated with mineral nanoparticles (17–41%) and in the free anions fraction (17–56%), and this latter likely includes As desorbed from mineral particles[29]. Similarly, much less soil As is extracted by water than Se (0.10 ± 0.07% of soil As against 1.1 ± 0.8% of soil Se; Supplementary Fig. 14). Figure 4c provides the distribution of As$_{water}$ among the SEC fractions, where the (organo)mineral colloids fraction includes the fraction "(nano)particles >20 nm" that does not elute out of our selected SEC columns. These As speciation data are only given for the Kohala topsoils because the filtration at 20 nm of water extracts and subsequent ICP-MS/MS analysis (Table 2), allowing estimation of As recovery by SEC, was only applied to these six soils. Nevertheless, the proportions of colloidal As >20 nm obtained for these six soils (12-66% of As$_{water}$; Fig. 4c) matched well with previous studies conducted on other sites[46,58] and are always higher than the proportions of colloidal Se >20 nm (0-19% of Se$_{water}$; Table 2). In these six soils, organic As

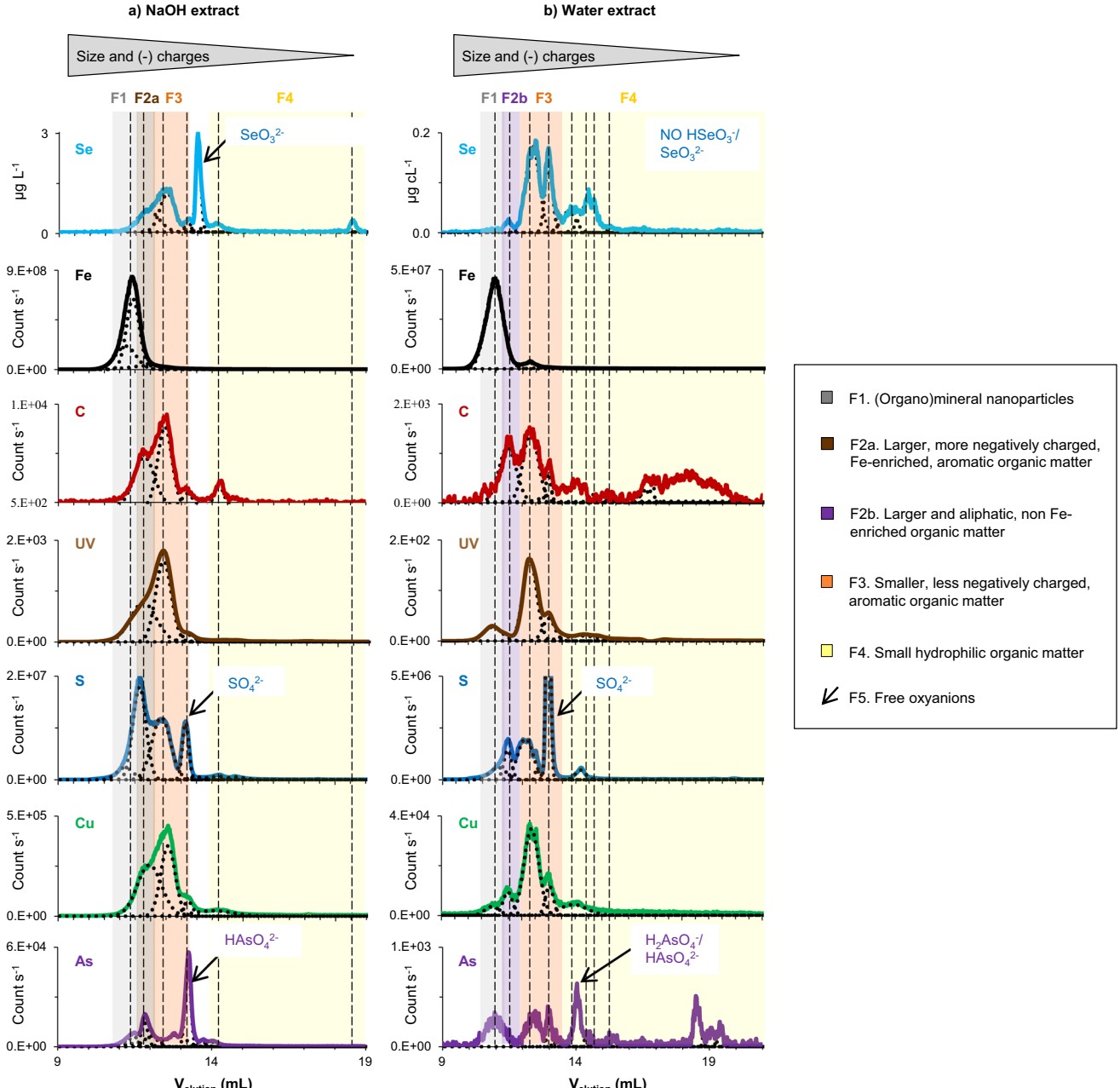

**Fig. 2 | Typical ultraviolet (UV) absorbance and element chromatograms obtained with the developed SEC-UV-ICP-MS/MS method for Kohala soil extracts together with their deconvolution.** Panel **a** shows the selenium (Se), iron (Fe), carbon (C), UV, sulfur (S), copper (Cu), and arsenic (As) chromatograms obtained for the NaOH extract of the topsoil (0–10 cm) from Kohala site S4 (intermediate soil organic carbon content), while panel **b** shows the same type of data for the ultrapure water extract of this soil. The optimized size exclusion chromatographic (SEC) separation involves two columns (Shodex OH-Pak 803-SB and 802.5-SB) connected in series and operated with a mobile phase containing 5 mmol L$^{-1}$ ammonium nitrate at pH 7 and 9.5 for ultrapure water and NaOH extracts, respectively. For Se, the mass flow chromatogram (unit, µg L$^{-1}$) resulting from the on-line isotope dilution is shown while the intensity chromatograms (unit, counts s$^{-1}$) are shown for the other elements. The background colors indicate the boundaries of the identified SEC fractions, except the free oxyanions (i.e., HSeO$_3^-$/SeO$_3^{2-}$, SO$_4^{2-}$ and H$_2$AsO$_4^-$/HAsO$_4^{2-}$) that are indicated with arrows. The vertical dashed lines show the apex of the identified fraction peak(s). Note that the fraction F2 corresponding to the first C peak is noted F2a for NaOH extracts and F2b for ultrapure water extracts because for F2a the peak of C coincides with an UV peak, while for F2b there is no UV peak associated with the C peak, therefore representing non-aromatic, aliphatic organic matter.

accounts for $32 \pm 13\%$ As$_{water}$ (Fig. 4d) against $75 \pm 26\%$ for Se$_{water}$, which together with the data for alkaline extracts, show a larger association of Se with SOM than As that is more bound to minerals.

Not only are the extents of As and Se association to SOM different but also their distribution among and within the different defined OM fractions. Organic As$_{NaOH}$ is more associated with larger, more hydrophobic and negatively charged, Fe-enriched, aromatic OM (F2a, $19 \pm 11\%$ against $9 \pm 7\%$ for Se$_{NaOH}$) than with smaller and/or less

hydrophobic aromatic OM (F3, $7 \pm 5\%$ against $9 \pm 7\%$; Figs. 4a and 3a). Even within F3, the Se peaks do not match those of As (Fig. 2), demonstrating that Se and As are associated with different organic compounds. The higher proportion of As$_{NaOH}$ associated to Fe-enriched OM is in line with previous XAS-based studies establishing the importance of ternary complexation for As binding to SOM via so-called Fe(III)-bridges[26]. Given the chemical similarities between Se and As oxyanions, this ternary complexation was previously proposed to

**Table 2 | SEC recoveries for selenium (Se), sulfur (S), copper (Cu), zinc (Zn), iron (Fe), arsenic (As), and lead (Pb) in NaOH extracts from Kohala topsoils (n = 6), and proportions of these elements in the <20 nm size fraction of ultrapure water and NaOH extracts from Kohala topsoils (n = 6)**

| Soil | Se | S | Cu | Zn | Fe | As | Pb |
|---|---|---|---|---|---|---|---|
| SEC recoveries in NaOH extracts (% of total extracted element)[a] | | | | | | | |
| S1 (0–10 cm; A) | 100 ± 11 | 109 ± 5 | 98 ± 2 | 89 ± 6 | 33 ± 1 | 85 ± 4 | 17 ± 2 |
| S2 (0–10 cm; A) | 102 ± 8 | 104 ± 4 | 109 ± 2 | 104 ± 18 | 60 ± 2 | 84 ± 5 | 56 ± 4 |
| S3 (0–10 cm; A) | 99 ± 16 | 108 ± 5 | 79 ± 1 | 88 ± 2 | 81 ± 1 | 85 ± 13 | 52 ± 2 |
| S4 (0–10 cm; A) | 102 ± 6 | 106 ± 7 | 108 ± 5 | 108 ± 3 | 75 ± 2 | 83 ± 6 | 55 ± 2 |
| S5 (0–10 cm; A) | 92 ± 9 | 105 ± 4 | 105 ± 1 | 94 ± 6 | 98 ± 11 | 81 ± 1 | 52 ± 1 |
| S6 (0–10 cm; A) | 91 ± 2 | 97 ± 2 | 102 ± 6 | 110 ± 5 | 81 ± 1 | 83 ± 6 | 54 ± 1 |
| Proportions of elements found in the <20 nm size fraction of NaOH extracts (in % of total extracted element)[b] | | | | | | | |
| S1 (0–10 cm; A) | 100 ± 4 | 110 ± 4 | 98 ± 1 | 71 ± 3 | 2.09 ± 0.04 | 75 ± 2 | 22 ± 1 |
| S2 (0–10 cm; A) | 91 ± 4 | 110 ± 2 | 100 ± 1 | 103 ± 13 | 20.6 ± 0.3 | 88 ± 1 | 40 ± 1 |
| S3 (0–10 cm; A) | 101 ± 4 | 103 ± 2 | 99 ± 1 | 35 ± 3 | 24 ± 1 | 83 ± 5 | 26 ± 1 |
| S4 (0–10 cm; A) | 93 ± 3 | 109 ± 5 | 100 ± 1 | 44 ± 3 | 19.6 ± 0.1 | 76 ± 1 | 28 ± 1 |
| S5 (0–10 cm; A) | 96 ± 2 | 93 ± 1 | 83 ± 1 | 31 ± 1 | 5.2 ± 0.1 | 89 ± 1 | 4.6 ± 0.1 |
| S6 (0–10 cm; A) | 99 ± 1 | 102 ± 1 | 102 ± 3 | 98 ± 1 | 99 ± 2 | 93 ± 4 | 58 ± 1 |
| Proportions of elements found in the <20 nm size fraction of water extracts (in % of total extracted element)[b] | | | | | | | |
| S1 (0–10 cm; A) | 69 ± 1 | 99 ± 3 | 47 ± 1 | 1.13 ± 0.02 | 0.08 ± 0.02 | 37 ± 2 | 1.3 ± 0.1 |
| S2 (0–10 cm; A) | 66 ± 4 | 97 ± 2 | 67 ± 1 | 3.3 ± 0.5 | 0.60 ± 0.01 | 53 ± 1 | 2.8 ± 0.1 |
| S3 (0–10 cm; A) | 95 ± 2 | 97 ± 1 | 82 ± 1 | 7.9 ± 0.3 | 1.47 ± 0.02 | 53 ± 3 | 2.0 ± 0.1 |
| S4 (0–10 cm; A) | 95 ± 1 | 95.3 ± 0.3 | 76 ± 1 | 12 ± 1 | 0.85 ± 0.01 | 34 ± 2 | 1.65 ± 0.04 |
| S5 (0–10 cm; A) | 99 ± 3 | 100 ± 6 | 98 ± 2 | 111 ± 1 | 19.4 ± 0.4 | 78 ± 6 | 9.6 ± 0.4 |
| S6 (0–10 cm; A) | 100 ± 2 | 94 ± 1 | 101 ± 2 | 114 ± 8 | 56 ± 1 | 88 ± 5 | 39 ± 1 |

[a]The SEC recoveries presented here were determined for each element by summing-up the concentrations measured by ICP-MS/MS in SEC fractions collected using an ISCO Foxy R R1 TELEDYNE fraction collector (with which maximal four fractions could be collected). For each element, the given standard deviation includes the standard deviations of the element concentration in each collected fractions and the standard deviation of the total element concentration in the extracts (with ICP-MS/MS acquisition being done in triplicate for all elements, fractions and soil extracts).
[b]The proportions of elements found in the size fraction <20 nm were determined after quantifying the elements in filtrates of NaOH and water extracts filtered at 20 nm, and are expressed in % of the element concentration in the NaOH and water extracts filtrated at 0.45 µm that are used for SEC-UV-ICP-MS/MS analysis. The standard deviations include the standard deviation obtained for the total elements concentrations in both the extracts filtered at 20 nm and at 0.45 µm (with ICP-MS/MS acquisition being done in triplicate for all elements and extract filtrates).

be an important mechanism for Se as well[16,24]. By showing that up to 11% of $Se_{NaOH}$ is associated with the Fe and As enriched OM fraction, our data suggests that Se ternary complexation may occur in soils, however, is not a dominant mechanism of Se association to SOM. Until now, one study has investigated the feasibility of ternary Se(IV)-Fe(III)-SOM complexation using a model system (i.e., Se(IV) incubation with reference humic substances pre-equilibrated with Fe(III))[24]. However, the SEC-ICP-MS method used in this earlier study was not resolutive enough to distinguish between these complexes and Se(IV) adsorbed to nanometer-sized Fe(III) (oxy)hydroxides that form in such model system[26,59].

### Se and S extractability are similar but their speciation strongly differs

Similarly to Se, a large share of Kohala soil S is extracted by NaOH (53 ± 12% versus 65 ± 12% of soil Se; Supplementary Fig. 13), and 1.8 ± 1.3% of soil S is extracted by water (versus 1.1 ± 0.8% of soil Se; Supplementary Fig. 14). Although their alkaline and water extractabilities are similar, the speciation of Se and S is different. $S_{NaOH}$ is more abundant in the larger, more hydrophobic and negatively charged, Fe-enriched, aromatic OM (F2a, 28 ± 17% of $S_{NaOH}$ against 9 ± 7% for $Se_{NaOH}$) and much less in the small hydrophilic OM (F4, 4 ± 2% of $S_{NaOH}$ against 19 ± 7% of $Se_{NaOH}$; Figs. 5a–3a). In water extracts, only 25 ± 13% of $S_{water}$ is organic, against 66 ± 21% for $Se_{water}$, and the dominant S compound is free sulfate ion ($SO_4^{2-}$(VI), 74 ± 14% of $S_{water}$; Fig. 5c, d) whereas free Se oxyanions account for only 13 ± 15% of $Se_{water}$. Within the water-soluble organic fractions, up to 20% of $S_{water}$ is associated with larger and aliphatic OM (against < 4% for $Se_{water}$), while only 7 ± 6% is found as small hydrophilic organic S (against 45 ± 17% for $Se_{water}$). Through demonstrating contrasts between Se and S organic

forms, and more generally their speciation in water and NaOH soil extracts, our results indicate that sources, formation and/or degradation pathways of soil organic Se and S may be significantly different.

Using XAS, soil organic S has been shown to occur in various moieties and oxidation states, i.e., thiols (C-SH), sulfide (C–S–C), sulfoxide (C-SO-C), sulfone (C–SO$_2$–C), sulfonate (C–SO$_3$), ester-sulfate (C–O–SO$_3$)[47]. Qin et al.[15] applied XAS to Se-contaminated soils (>10 mg(Se) kg$^{-1}$) and concluded that organic Se involves Se(-II)-C binding. However, the organic Se moieties and oxidation state still remain unsure, as the only organic Se standards commercially available for XAS spectra fitting are Se(-II, -I)-C containing compounds. By applying SEC-UV-ICP-MS/MS to soil extracts, our study provides a comparison between organic Se and S in term of their overall structure (i.e., size, negative charge, and aromaticity). Notably, it highlights large proportions of organic S but only low proportions of organic Se associated to the larger and aliphatic OM fraction (F2b) of water extracts at the (sub)humid Kohala sites S3-6. This aliphatic organic S fraction most probably includes aliphatic sulfonates, sulfolipids and/or other aliphatic S compounds produced by bacteria and/or plants[60,61], for which Se-analogs were to the best of our knowledge never identified. By combining the speciation data for Kohala soils with data on Se and S concentrations in dominant plant species collected at Kohala study sites (Supplementary Table 5), we further observed that the ratios of organic $S_{NaOH}$ to organic $Se_{NaOH}$ in soils are always much lower than the S to Se ratio in plant roots and leaves (Supplementary Fig. 15). Moreover, S concentrations are higher in leaves than in mixed roots whereas the opposite is found for Se (Supplementary Fig. 16). These results suggest that organic Se is mostly formed in soils in contrast to organic S that includes an important pool coming from plants that

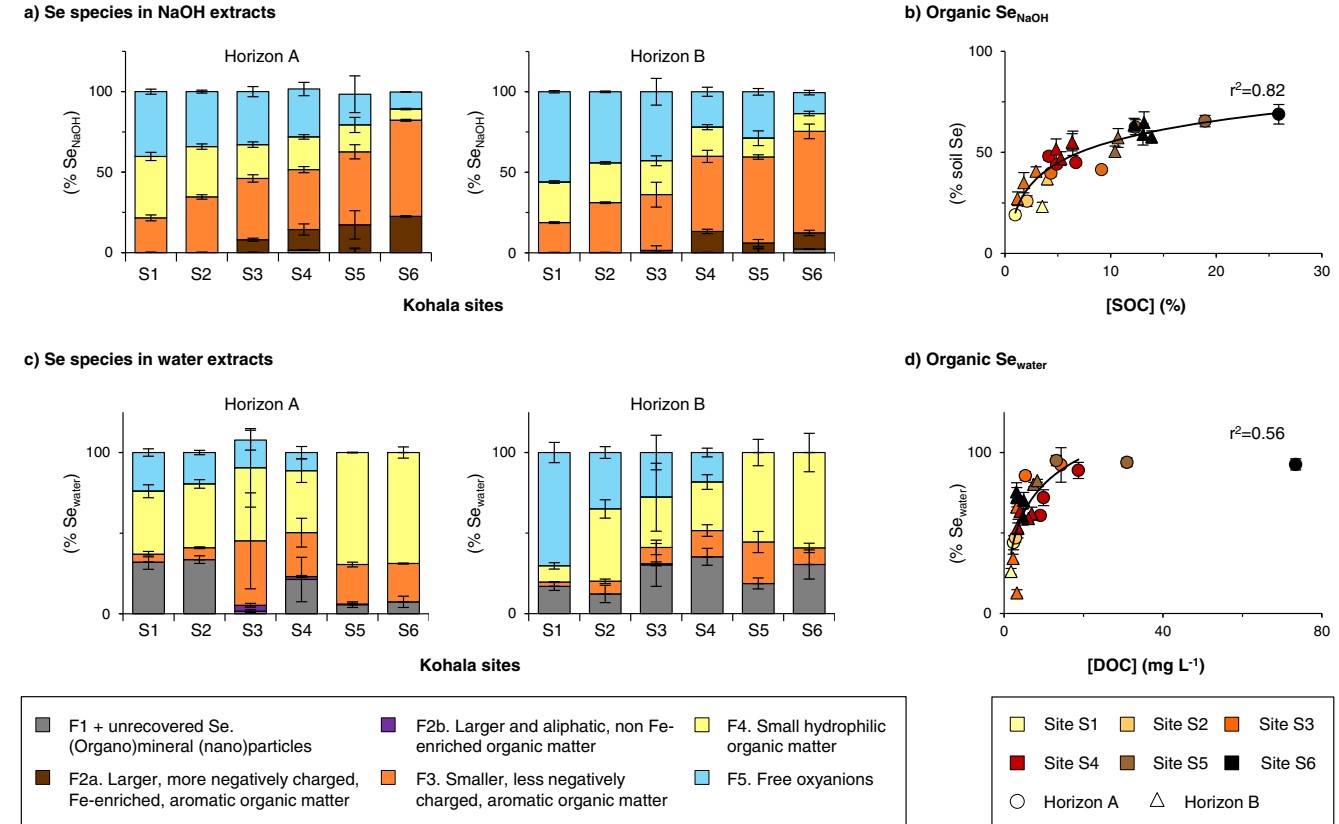

**Fig. 3 | Speciation of selenium (Se) in NaOH and ultrapure water extracts of Kohala soils along the rainfall gradient.** Panel **a** provides the distribution of Se among the five identified size exclusion chromatography (SEC) fractions in NaOH extracts of soil horizons A and B. Panel **b** shows the proportion of NaOH-extractable organic Se (sum of SEC fractions F2–F4) in all analyzed soils ($n = 25$) as a function of soil organic carbon (SOC) concentrations. Panel **c** provides the distribution of Se among the five identified SEC fractions in ultrapure water extracts of soil horizons A and B, where the proportions of water-soluble Se unrecovered by SEC, likely corresponding to Se associated to >20 nm colloids, was added to SEC fraction F1. Panel

**d** shows the proportion of water-soluble organic Se (sum of SEC fractions F2–F4) in all analyzed soils ($n = 25$) as a function of dissolved organic carbon (DOC) concentrations. Correspondence between soil horizons and soil depths is given in Supplementary Table 1. For all presented data, the error bars represent standard deviations, which consider (i) the standard deviation obtained during the SEC peak deconvolution; (ii) the standard deviation of soil Se concentrations that were determined in triplicate (for data in panels **b** and **d**); and (iii) the standard deviation resulting from averaging the measured data for soil depth samples to obtain the data for soil A and B horizons (for data in panels **a** and **c**).

---

actively take up airborne S, which is transformed to organic S within the plant[62].

## Soil enrichment of Se, As and S explained by their speciation

Insights into soil Se accumulation can be obtained by looking at the enrichment factors with respect to parent rock (EFs) of total Se and NaOH-extracted Se species in soil horizon A along the Kohala rainfall gradient, and by comparing them to enrichment of As and S (Fig. 6). Kohala soil parent rock contains $0.140 \pm 0.005$ mg(Se) kg$^{-1}$, $3.1 \pm 0.1$ mg(As) kg$^{-1}$ and 0.06 g(S) kg$^{-1}$, and the EFs of soil Se, As and S resulting from both soil development processes and atmospheric inputs[41,63] are 112–1358%, 54–326%, and 343–4237%, respectively (Fig. 6a-c). Atmospheric inputs are not a focus of our study, but rainfall and dry atmospheric depositions derived from sea-salt, volcanic dust and/or Asian dust are known to dominate S inputs in Kohala soils[64]. It is likely that atmospheric depositions are also important Se sources to these soils[5], given that soil Se EFs values are between those of S and Pb (for which the atmosphere is a main source to Kohala soils[63,64]) and those of Fe, titanium, and manganese (for which parent rock weathering dominates the inputs[63]; Supplementary Fig. 17).

Along the rainfall and SOC gradient, the EF of soil Se and total organic Se$_{NaOH}$ increases, except at site S6 (site of strong leaching[40,41,43]) where it decreases by 2 times compared to site S5 (Fig. 6a). In contrast, the EF of mineral-adsorbed Se(IV) (estimated from Se(IV)$_{NaOH}$[29,30] remains stable between S3 and S5 despite

increasing content of amorphous Fe (oxy)hydroxides (Fig. 1b) that are often considered as a key controlling phase of soil Se accumulation (especially in volcanic soils[9,10]). In addition, the EF of soil Se that is not extracted by NaOH, which likely includes Se strongly sorbed to minerals, Se(0) nanoparticles, metal Se(-II)[30] and non-extractable organic Se, decrease from S3 to S5 (Fig. 6a). These results clearly shows that SOM increasingly drives soil Se accumulation along the climosequence with Se association to alkaline-extractable SOM explaining ~50 up to 70% of Se accumulation in soils containing 5–7% (sites S3-S4) up to 16–26% (sites S5-S6) of SOC. Our data further indicates that the loss of soil Se at site S6 results mainly from a loss of mineral-adsorbed Se(IV) and small hydrophilic organic Se, and to a lesser extent of larger, more negatively charged and hydrophobic organic Se. Indeed, the EF of small hydrophilic Se and Se(IV)$_{NaOH}$ decrease 4 times between S5 and S6, while the EF of the other organic Se fractions decrease 1.3–1.7 times (Fig. 6d). Mineral-adsorbed Se(IV) is probably desorbed due to the intense leaching occurring at this site and the reductive dissolution of (oxy)hydroxides occurring during heavy rainfall[40,41]. Interestingly, at site S6, the loss of soil As that is more bound to minerals (see As$_{NaOH}$ bound to mineral colloids, As(V)$_{NaOH}$, and As non-extracted by NaOH[29]) is two times greater than for Se (Fig. 6a, b). The contrasting forms between soil Se and S also explain the lower Se then S accumulation at site S6. Whereas small hydrophilic organic Se and mineral-adsorbed Se(IV) can be leached out (Fig. 6d), S that is largely present as larger and/or more hydrophobic organic

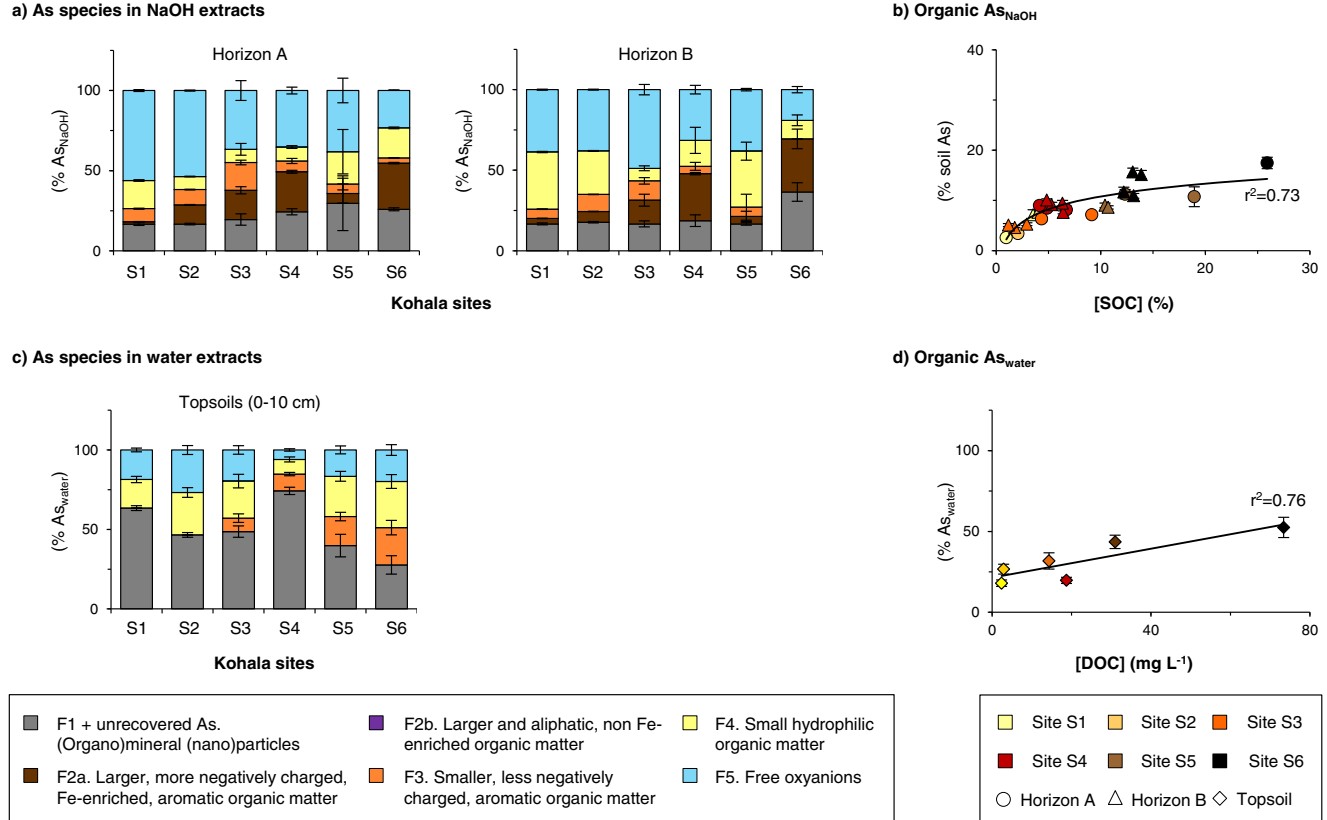

**Fig. 4 | Speciation of arsenic (As) in NaOH and ultrapure water extracts of Kohala soils along the rainfall gradient.** Panel **a** provides the distribution of As among the five identified size exclusion chromatography (SEC) fractions in NaOH extracts of soil horizons A and B. Panel **b** shows the proportions of NaOH-extractable organic As (sum of SEC fractions F2–F4) in all analyzed soils ($n = 25$) as a function of soil organic carbon (SOC) concentrations. Panel **c** provides the distribution of As among the >20 nm colloids fraction (determined by As quantification in extracts filtered at 20 nm) and the five identified SEC fractions in ultrapure water extracts of the six Kohala topsoils (0–10 cm). Panel **d** shows the proportions of water-soluble organic As (sum of SEC fractions F2–F4) in the six topsoils as a function of dissolved organic carbon (DOC) concentrations. Correspondence between soil horizons and soil depths is given in Supplementary Table 1. For all presented data, the error bars represent standard deviations, which consider i) the standard deviation obtained during the SEC peak deconvolution; ii) the standard deviation of soil As concentrations that were determined in triplicate (for data in panels **b** and **d**); and iii) the standard deviation resulting from averaging the measured data for soil depth samples to obtain the data for soil A and B horizons (for data in panels **a** and **c**).

compounds appears less leachable (Fig. 6c, f). Our data thus show that although Se association to SOM overall immobilizes Se thereby preventing this micronutrient from being leached out, the capacity of SOM to retain Se depends on Se distribution among the different chemical and size organic fractions.

## Relationships between Se and S concentrations in plants and their water-soluble speciation

To get insights into plant Se availability, we related Se and S speciation in water extracts with Se and S concentrations in leaves from the dominant plant species and in a mixture of roots at each of the study sites. First, plant Se and S concentrations are much higher at the nutrient-uplifting site S3 than at the other five sites (Supplementary Fig. 16). For example, grass leaves contain $1.5 \pm 0.2$ mg(Se) kg$^{-1}$ and $2.7 \pm 0.2$ g(S) kg$^{-1}$ at site S3 versus <0.2 mg(Se) kg$^{-1}$ and <2.7 ± 0.2 g(S) kg$^{-1}$ at the other sites. Site S3 is however a specific Kohala site where nutrients are enriched in topsoils with respect to deeper soils, and the water balance probably favors the plant-uptake of nutrients (precipitation is very close to evapotranspiration)[41,44]. Aside from this peculiar site, the lowest plant Se concentrations are found at sites S5-S6 (average, $0.080 \pm 0.008$ and $0.15 \pm 0.03$ mg kg$^{-1}$ in grass leaves and mixed roots, respectively; Supplementary Fig. 16a) where Se$_{water}$ is dominated by organic forms and does not include free Se oxyanions (Fig. 3c). Compared to sites S5-S6, plant Se concentrations are two times higher at S1, S2 and S4 (average, $0.19 \pm 0.04$ and

$0.29 \pm 0.05$ mg kg$^{-1}$ in grass leaves and mixed roots) where free Se oxyanions accounts for 5–26% (horizon A) and 16-57% (horizon B) of Se$_{water}$ (Fig. 3c). In contrast to Se, the concentrations of S in grass leaves and mixed roots are twice higher at (sub)humid sites (average S4-S6, $1.93 \pm 0.05$ and $1.3 \pm 0.1$ g kg$^{-1}$, respectively) than at arid sites (average S1–S2, $1.06 \pm 0.06$ and $0.78 \pm 0.01$ g kg$^{-1}$, respectively; Supplementary Fig. 16b). When excluding the nutrient-uplifting site S3, the concentrations of Se oxyanions in topsoils (0-10 cm, corresponding to the penetration depth of plant roots analyzed; data shown in Supplementary Fig. 18) positively correlate with the Se concentrations in plant leaves and roots ($p < 0.05$; Fig. 7a). Similarly, the water-soluble sulfate concentrations positively correlate with the S concentrations in plant roots ($p < 0.05$), and there is a positive trend between the water-soluble sulfate concentrations and the S concentrations in plant leaves ($p = 0.19$; Fig. 7b). From these correlations and the more prominent fraction of S than Se found as free oxyanions (Figs. 3c and 5c), we infer that the high proportion of organic Se in the water-soluble phase, although mainly composed of small hydrophilic compounds (other than plant-available Se-amino acids), limits Se availability for plants.

In contrast to Se for which there is a negative trend between the concentrations of organic Se fractions and the Se concentrations in plant leaves and roots ($p = 0.19$; Fig. 7c), the concentrations of organic S in larger and aliphatic OM and smaller and/or less hydrophobic, aromatic OM positively correlate with plant S concentrations (Fig. 7b, d). This could be explained by plants being an important source of

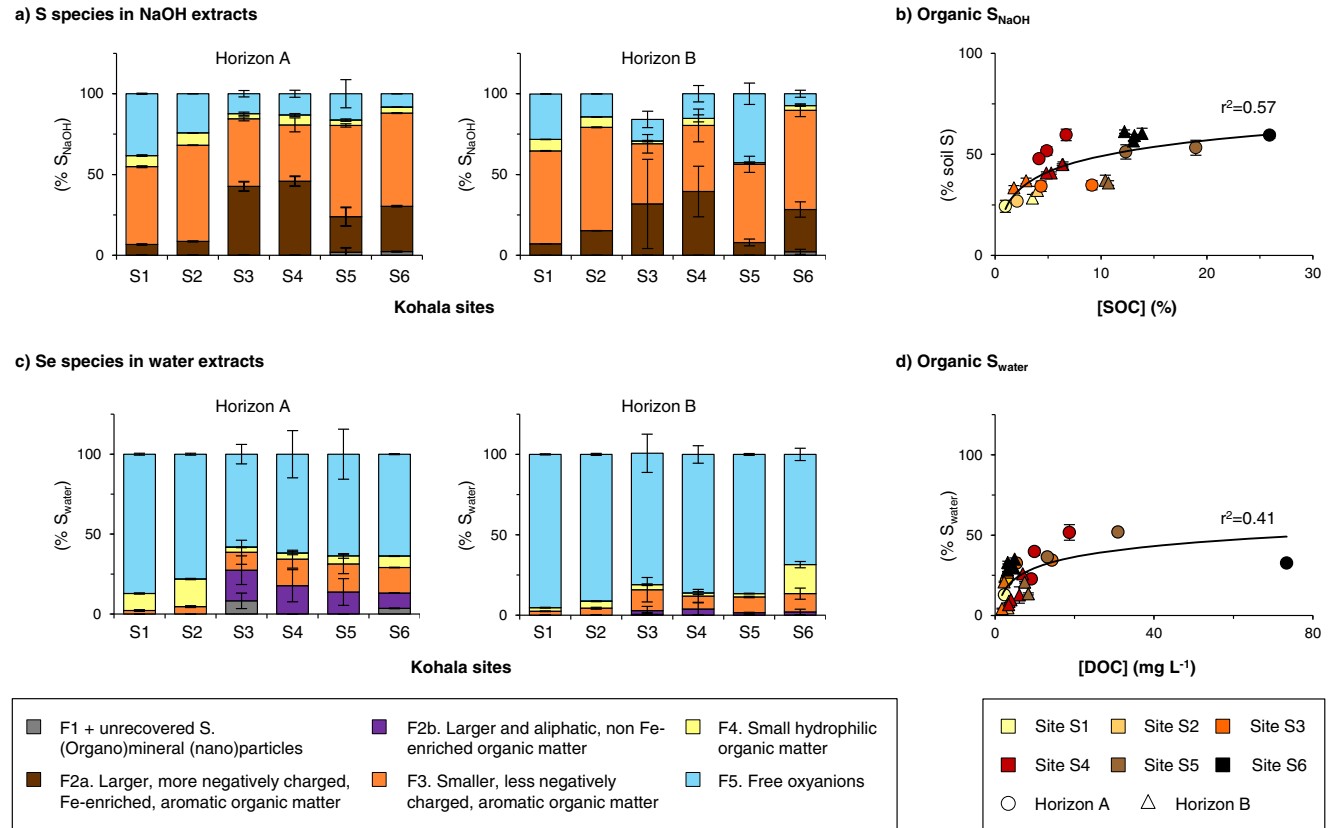

**Fig. 5 | Speciation of sulfur (S) in NaOH and ultrapure water extracts of Kohala soils along the rainfall gradient.** Panel **a** provides the distribution of S among the five identified size exclusion chromatography (SEC) fractions in NaOH extracts of soil horizons A and B. Panel **b** shows the proportion of NaOH-extractable organic S (sum of SEC fractions F2-F4) in all analyzed soils ($n = 25$) as a function of soil organic carbon (SOC) concentrations. Panel **c** provides the distribution of S among the five identified SEC fractions in ultrapure water extracts of soil horizons A and B. Panel **d** shows the proportion of water-soluble organic S (sum of SEC fractions F2–F4) in all analyzed soils ($n = 25$) as a function of dissolved organic carbon (DOC) concentrations. Correspondence between soil horizons and soil depths is given in Supplementary Table 1. For all presented data, the error bars represent standard deviations, which consider i) the standard deviation obtained during the SEC peak deconvolution; ii) the standard deviation of soil S concentrations that were determined in triplicate (for data in panels **b** and **d**); and iii) the standard deviation resulting from averaging the measured data for soil depth samples to obtain the data for soil A and B horizons (for data in panels **a** and **c**).

organic S (but not Se) to soils via litterfall and root deposit after internalization of gaseous S in organic S[62], but also by the mineralization of organic S into sulfate, which is an essential process for S acquisition by plants[61,65]. Mineralization of water-soluble organic Se into Se oxyanions was previously proposed to be an important process in Se uptake by plants as well[11,20–22], but our data suggest a less efficient mineralization of both large and small organic Se forms into Se oxyanions as compared to S. Whether this is true or not and is related to the intrinsic chemical properties of Se species or soil enzymatic activities calls for further studies to investigate the processes of organic Se formation and degradation in soils in more details.

The SEC-UV-ICP-MS/MS method presented in this study offers a window into the black-box approach of soil extractions that remains central to assess the speciation of Se and other elements in soils. As the necessary instrumentation is relatively easily accessible compared to advanced solid-phase techniques such as XAS, this method will enable the investigation of large collections of soils and their pore waters as well as the study of time-resolved experiments performed with natural soils. Furthermore, our SEC method can be combined with high-resolution mass spectrometry after appropriate sample or fraction pre-concentration. This will allow structural elucidation of organic forms of trace elements and thus better inform on the mechanisms for trace element binding or incorporation into SOM as well as on the degradation and plant availability of the organic-trace element complexes or compounds. More specifically for Se, an important next step is to test the positive correlations observed between the alkaline-

extractable organic Se and SOC at larger spatial-scale and across agricultural systems. Such statistical relationships may indeed offer the possibility to model soil Se speciation and plant availability, which are of key importance to predict crop Se contents and future trends in Se status in food-crops with changing climate and soil management[3,66]. Although we prove that multi-element speciation in soil extracts is feasible, optimizations are still needed for cationic trace elements (e.g., $Zn^{2+}$ or $Cu^{2+}$) to extract their organic fractions without potential re-precipitation (e.g., into $ZnO_{(s)}$) occurring during alkaline extraction and to elute their ionic forms. Using SEC columns specific for nanoparticles[46] or field flow fractionation[45] would also be important to characterize the size fractions >20-40 nm of water-soluble Se and As. By providing a blueprint for developing multi-element speciation datasets, our SEC method represents an important step towards a better understanding of micronutrient availability and uptake, which are critical information to improve nutrition and combatting hidden hunger.

## Methods

### Sample collection and preparation
The soil and plant (leaves and mixed roots) samples were collected in February 2015, and a parent rock sample was hammered out of an exposed bedrock near site S1 in March 2017. Soil samples were air-dried and gently ground using a mortar before sieving to <2 mm using a Retsch sieve. The sieve and mortar were subsequently rinsed with ultrapure water, ethanol and dried between samples. The <2 mm

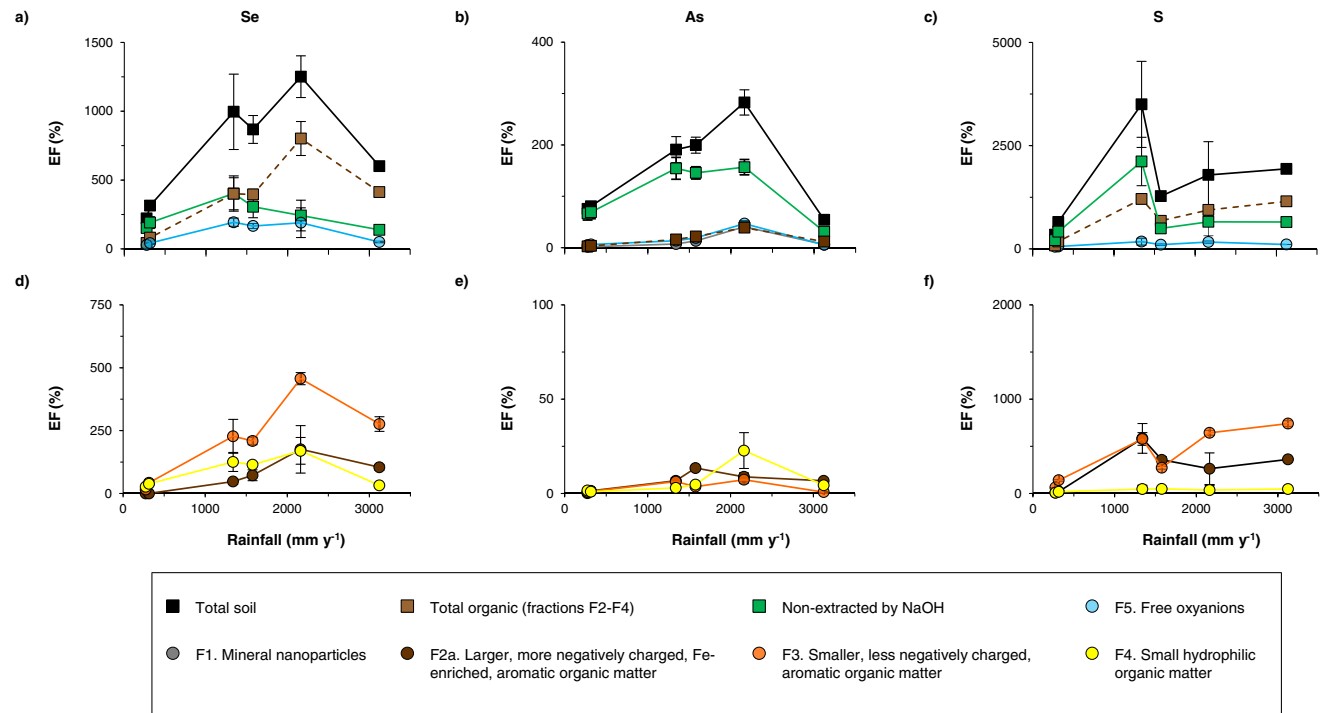

**Fig. 6 | Enrichment factors (EFs) with respect to parent rock of total soil selenium (Se), arsenic (As), and sulfur (S) and of NaOH-extracted Se, As, and S species in soil horizon A along the Kohala rainfall gradient.** Panel **a** provides the EFs of total soil Se as well as of total organic $Se_{NaOH}$, $Se_{NaOH}$ oxyanions, $Se_{NaOH}$ associated to mineral nanoparticles and Se non-extracted by NaOH in soil horizon A. Similarly, panel **b** provides the EFs of total soil As as well as of total organic $As_{NaOH}$, $As_{NaOH}$ oxyanions, $As_{NaOH}$ associated to mineral (nano)particles and As non-extracted by NaOH in soil horizon A. Panel **c** provides the EFs of total S as well as of total organic $S_{NaOH}$, $S_{NaOH}$ oxyanions, and S non-extracted by NaOH in soil horizon A. Panels **d**–**f** show the EFs of the different organic $Se_{NaOH}$, $As_{NaOH}$ and $S_{NaOH}$ fractions in soil horizon A, respectively. Note that an EF value below and above 100% indicate a loss and accumulation of the element with respect to parent rock, respectively. For all presented data, the error bars represent standard deviations, which consider i) the standard deviation of soil Se, As and S concentrations that were determined in triplicate (for data in panels **a**–**c** only) or the standard deviation obtained during the SEC peak deconvolution (for data in panels **d**–**f** only); ii) the standard deviation of parent rock Se, As and S concentrations that were determined in triplicate; and iii) the standard deviation resulting from averaging the measured data for soil depth samples to obtain the data for soil A horizons.

fraction was then ground using a disc swing mill (RS1, Retsch). The mixed roots were carefully cleaned in 6 mmol L$^{-1}$ ammonium sulfate (trace metal analysis; Sigma-Aldrich) at a temperature of 4 °C in six subsequent baths. The cleaned roots and plant leaves were then freeze-dried and ground with the disc swing mill. The parent rock sample was broken using a hammer and ground with the disc swing mill (RS1, Retsch).

### Determination of soil properties

Soil pH was measured in 10 mmol L$^{-1}$ CaCl$_2$ (soil:solution ratio 1:2.5) after 2 h of equilibration using a combined pH electrode. Total carbon (TC) and nitrogen (TN) contents were determined using a CNS analyzer (Euro EA 3000; Eurovectors SPA) and total inorganic carbon (TIC) was measured with a Coulomat (CM 5015 Coulometer; UIC Inc). Total organic carbon (TOC) was calculated as the difference between TC and TIC. The estimation of amorphous and crystalline Fe (oxy)hydroxides by oxalate and dithionite citrate bicarbonate extractions and subsequent Fe quantification in the extracts was taken from Helfenstein et al.[67].

### Element quantification in soils, rock and plant leaves and roots

**Soils and rock.** S and niobium (Nb) were quantified by X-Ray Fluorescence spectroscopy (Spectro, XEPOS) on pellets prepared with 4 g of grounded soil or rock mixed with 0.9 g of wax (Cereox, Licowax). Se, As, Fe, Cu, Zn, and Pb were quantified by ICP-MS after microwave-assisted acid digestion (MLS GmbH, UltraCLAVE 4). Fifty mg of soil or parent rock was digested in triplicate with 1 mL of H$_2$O$_2$ (30%, for ultratrace analysis; Sigma-Aldrich), 5 mL of

HNO$_3$ (65%, Suprapur; Merck) and 0.3 mL of HF (48%; ≥99.99% trace metal basis; Merck). The microwave program was as follows: temperature ramp from 25 to 230 °C over 25 min (Power, 2500 W; P, 130 Bar) and then, 20 min at 230 °C (Power, 2500 W; P, 130 Bar). The polytetrafluoroethylene (PTFE) microwave vessels were first cleaned using 30% HNO$_3$ (65%, Suprapur; Merck) using the same microwave-digestion program. After digestion, the digests were filled up to 50 mL with ultrapure water, then filtered with 0.45 μm syringe filter (Perfect-Flow®, Nylon membrane), and stored at 4 °C until elemental quantification. Elements were quantified in the digests using an Agilent 7500cx ICP-MS equipped with an octopole collision/reaction cell (C/RC), a concentric nebulizer, a Scott double-pass spray chamber cooled to 2 °C, and nickel sampler and skimmer cones. All ICP-MS parameters were optimized using a tuning solution containing 10 μg L$^{-1}$ of lithium (Li), yttrium (Y), cobalt (Co), cerium (Ce), and tellurium (Te) (prepared with standards from J.T. Baker). As ($m/z$ 75), and Se ($m/z$ 78) were measured with 5 mL min$^{-1}$ H$_2$ in the C/RC while Fe ($m/z$ 56), Cu ($m/z$ 63), and Zn ($m/z$ 66) with 4.5 mL min$^{-1}$ He, and Pb ($m/z$ 208) without gas in the C/RC. Acquisition parameters were: 0.1 s integration time and 3 replicates. Quantification was done by external calibration with elemental standards (purchased at J.T. Baker) prepared in the digestion matrix, and an internal standard containing scandium (Sc; 70 μg L$^{-1}$), indium (In; 7 μg L$^{-1}$) and lutetium (Lu; 7 μg L$^{-1}$) was used to check signal stability during the runs. Four certified reference materials (i.e., the basalt rock NIST SRM 688 and the soils Sigma-Aldrich CRM-044, Chinese Academy of Geological Science GBW 07405, and NIST SRM2709a) were

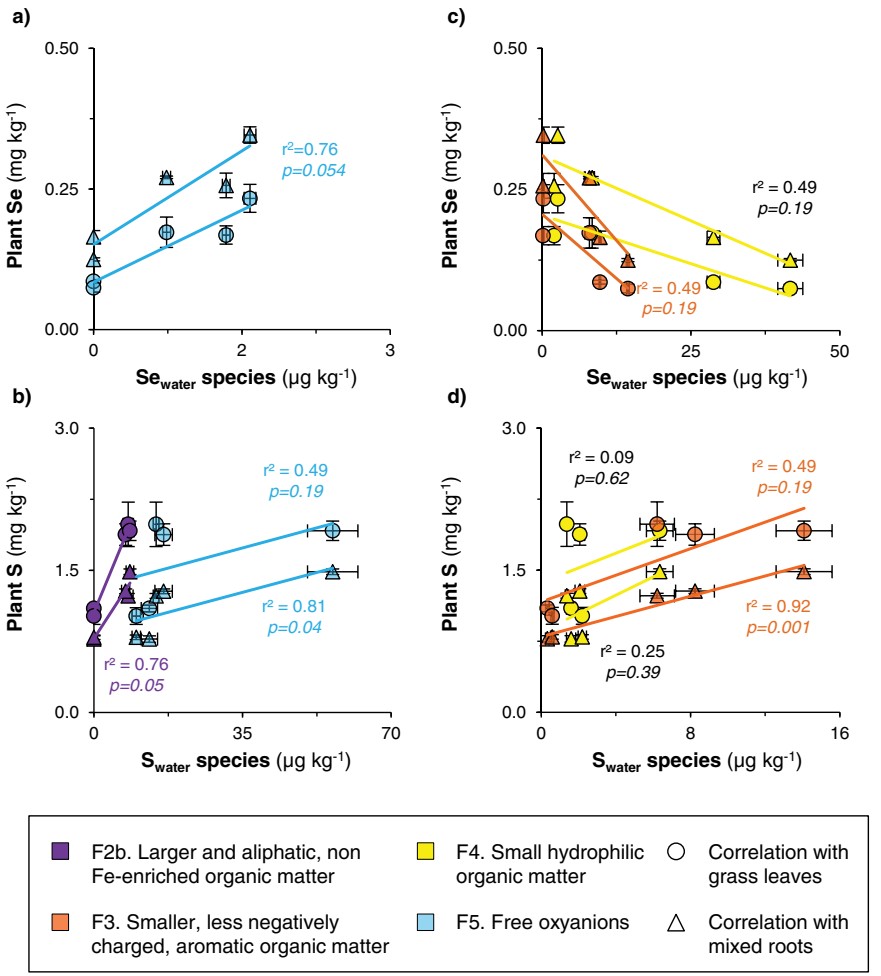

**Fig. 7 | Relationships of the concentrations of selenium (Se) or sulfur (S) in grass leaves and mixed roots with the water-soluble Se or S speciation in Kohala topsoils (0–10 cm), when excluding the site S3.** Panel **a** shows the correlations between the concentrations of Se in plants and the concentrations of Se oxyanions in ultrapure water extracts. Similarly, panel **b** shows the correlations between the concentrations of S in plants and the concentrations of sulfate or of larger and aliphatic organic S in ultrapure water extracts. Panel **c** shows the correlations between the concentrations of Se in plants and the concentrations of smaller, less negatively charged aromatic Se and small hydrophilic Se in ultrapure water extracts. Similarly, panel **d** shows the correlations between the concentrations of S

in plants and the concentrations of less negatively charged aromatic S and small hydrophilic S in ultrapure water extracts. The correlation coefficient ($r^2$) and associated p values were determined using 2-tailed Spearman correlation. Note that the correlations between the concentrations of total Se in grass leaves and mixed roots and the concentrations of water-soluble Se associated to (organo)mineral (nano)particles is given in Supplementary Fig. 19. For all presented data, the error bars represent standard deviations, which consider i) the standard deviation of plant Se, As and S concentrations that were determined in triplicate; and ii) the standard deviation obtained during the SEC peak deconvolution.

digested in triplicate and analyzed with the samples. Recoveries were between 96–106% and 88-110% for Se and the other elements, respectively (Supplementary Table 6 in Supplementary Method 1).

**Plant samples.** Plant leaves (as described in Supplementary Table 5) and roots were digested in triplicate by microwave-assisted acid digestion with the same program used for soil and rock samples, except that a mixture of 2 mL $HNO_3$ (65%, Suprapur; Merck) and 2 mL $H_2O_2$ (30%, for ultratrace analysis; Sigma-Aldrich) was used. After digestion, the digests were filled up to 10 mL with ultrapure water, then filtered with 0.45 μm syringe filter (Perfect-Flow®, Nylon membrane), and stored at 4 °C until elemental quantification. Se, As, and S were quantified in the digests using an Agilent 8900 ICP-MS/MS equipped with a concentric nebulizer, a Scott double-pass spray chamber cooled to 2 °C, a high-throughput injection system (ISIS) with a PTFE sample loop and platinum sampler and skimmer cones. All ICP-MS/MS parameters were optimized using a tuning solution as previously described. Se ($m/z$ 78 to >78) and As ($m/z$ 75 to >75) were measured in MS/MS

mode with 5 mL min⁻¹ $H_2$ in the C/RC, and S ($m/z$ 32 to >48) in MS/MS mode with 5 mL min⁻¹ $H_2$ and 30% $O_2$. Acquisition parameters were: 0.05–0.3 ms integration time (depending on the element) and 3 replicates. Quantification was done by external calibration prepared in the digestion matrix, and an internal standard was used to check signal stability as described in the subsection above about the analysis of soil and rock digests. The data treatment was performed using MassHunter software (Agilent). Two plant certified reference materials (i.e., NIST SRM1515 and NCS DC 73349) were digested in triplicate and analyzed with the samples. Recoveries between 96-97% and 80-88% were obtained for Se and other elements, respectively (Supplementary Table 7 in Supplementary Method 1).

**Soil extractions and elemental quantification in the extracts**

Soil extractions with ultrapure water (18.2 MΩ cm; Nanopure DIamond™ system) and NaOH (0.1 mol L⁻¹, Emsure, ≥99.0%; Merck) were performed following the method of Tolu et al.[13]. The mixture (solid:liquid ratio, 1:33) was shaken at 250 rpm during 24 h (room temperature) and centrifuged at 2'700 g (10 min). The

supernatant was filtered with 0.45 μm syringe filter (Perfect-Flow®, Nylon membrane) and stored at 4 °C until elemental quantification (within two weeks) and speciation analysis (within max. 2 days). Elemental quantification was done using the same Agilent 8900 ICP-MS/MS and the same tuning, internal standard and data treatment procedures as described for the analysis of plant digests. Se ($m/z$ 78 to >78) and As ($m/z$ 75 to >75) were measured in MS/MS mode with 5 mL min$^{-1}$ H$_2$ in the C/RC, Fe ($m/z$ 56 to >56), Cu ($m/z$ 63 to >63) and Zn ($m/z$ 66 to >66) in single-quadrupole mode with 4.5 mL min$^{-1}$ He, S ($m/z$ 32 to >48) in MS/MS mode with 5 mL min$^{-1}$ H$_2$ and 30% O$_2$, and Pb (208) in single-quadrupole mode without C/RC gas. Acquisition parameters were: 0.05–0.3 ms integration time (depending on element) and 3 replicates. Quantification was done by external calibration in the corresponding matrices (diluted extraction solution) and the absence of matrix effects was verified with standard additions using extracts of three contrasting Kohala topsoils. For each run, two freshwater certified reference materials (i.e., NIST1643f and National Water Research Institute, Canada, TMDA-51.2) diluted 10 and 100-fold in the extract matrices were analyzed with the samples, and recoveries between 96–106% and 85–116% were obtained for Se and other elements, respectively (Supplementary Table 8 in Supplementary Method 1).

For the water extracts, the S:L ratio was, in a second step, increased to 1:5 (a ratio often used to extract the water-soluble TE) to obtain SEC-UV-ICP-MS/MS signal for certain elements like C or As. Although higher extraction efficiencies were obtained when using the initial 1:33 ratio, both S:L ratio depicted the same trend in extraction efficiencies for Se in Kohala soils (Supplementary Fig. 20 in Supplementary Method 2). For all extracts, dissolved organic carbon (DOC) was measured using a TOC analyzer (Shimadzu TOC-L CSH).

## Se, S and As speciation in soil extracts by SEC-UV-ICP-MS/MS

Size exclusion chromatography (SEC) was performed by coupling an Agilent 1260 Infinity II high performance liquid chromatography (HPLC) system to an Agilent diode array detector (DAD) connected to the Agilent 8900 ICP-MS/MS described previously. First, the SEC separation was optimized using ultrapure water and NaOH extracts from three contrasting Kohala topsoils (0–10 cm) by testing different SEC columns and mobile phase composition (cf. Supplementary Table 3 in Supplementary Discussion 1, which provides all operating conditions used for the optimization of the SEC separation). The optimal SEC method was then selected based on chromatographic resolution of the element separation and recovery of Se species (Supplementary Discussion 2). Secondly, the analysis of all soil extracts was carried out using the optimal SEC method, which involves a series of Shodex OH-pak SB-803 and −802.5 HQ columns and a mobile phase containing 5 mmol L$^{-1}$ ammonium nitrate at pH 7.5 and pH 9 for water and NaOH extracts, respectively (cf. Supplementary Table 3 in Supplementary Discussion 1, which provides all operating conditions used for the optimal SEC method).

For both the optimization phases and the final analysis of all Kohala soil extracts, the mobile phase flow was of 1 mL min$^{-1}$, the sample injection volume of 100 μL, the UV absorbance was detected at a wavelength of 254 nm, and the Agilent 8900 ICP-MS/MS system and its tuning were as described for the analysis of digests. Each Kohala soil extract was measured by SEC-UV-ICP-MS/MS twice using different gases in the C/RC to obtain data on Se together with various elements. In the first SEC-UV-ICP-MS/MS run, 5 mL min$^{-1}$ H$_2$ was used in the C/RC to detect Se ($m/z$ 77 to >77, 78->78 and 80 to >80; 0.3 ms integration time) together with As ($m/z$ 75 to >75; 0.3 ms), Fe ($m/z$ 56 to >56; 0.1 ms), Cu ($m/z$ 65 to >65; 0.1 ms), Zn ($m/z$ 66 to >66; 0.1 ms) and Pb ($m/z$ 208 to >208; 0.1 ms). In the second SEC-UV-ICP-MS/MS run, 25% O$_2$ and 1 mL min$^{-1}$ H$_2$ was used in the C/RC to detect Se ($m/z$ 77 to >93, 78 to >94 and 80 to >96; 0.3 ms integration time) together with S ($m/z$

32 to >48; 0.1 ms), C ($m/z$ 12 to >28; 0.3 ms) and Ti ($m/z$ 47 to >63; 0.1 ms). For each SEC-UV-ICP-MS/MS run, internal standards were added post-UV detector (as detailed in Supplementary Table 3) to account for change in sensitivity during the SEC optimization phases (Supplementary Discussion 2) or to check for no significant sensitivity loss during the final analysis of all Kohala soil extracts (Supplementary Discussion 5).

For the final analysis of all Kohala soil extracts only, a $^{78}$SeO$_3^{2-}$(IV) standard was added post-UV detector (together with the internal standards) to quantify Se in SEC peaks by on-line Se isotope dilution (ID), after adapting the on-line ID calculations of Sariego Muñiz et al.[68] to the modified $^{80}$Se/$^{78}$Se isotopic ratio. The Se mass flow chromatograms resulting from the on-line ID calculations were then deconvoluted to obtain the Se amount in each SEC peak, from which the Se concentrations is obtained considering the sample injection volume (i.e., 100 μL). The $^{78}$SeO$_3^{2-}$(IV) standard (stock solution, 1000 mg L$^{-1}$) was prepared from an $^{78}$Se(0) standard (Isoflex, USA) according to the procedure of Dael et al.[69], and was then stored at 4 °C. This standard was composed of 0.0063 ± 0.0001% of $^{74}$Se, 0.003 ± 0.001% of $^{76}$Se, 0.16 ± 0.04% of $^{77}$Se, 99.50 ± 0.02% of $^{78}$Se, 0.33 ± 0.01% of $^{80}$Se and 0.008 ± 0.004% of $^{82}$Se (determined by ICP-MS/MS), and only consisted of SeO$_3^{2-}$(IV) (determined by AEC-ICP-MS/MS). The on-line isotope dilution calculation is given in Supplementary Method 3. The concentration of $^{78}$SeO$_3^{2-}$(IV) added post-UV detector was adjusted for all samples to reach an optimal $^{80}$Se/$^{78}$Se ratio during the SEC-elution. This is essential to obtain minor error propagation during on-line ID calculation and thus accurate Se quantification in the SEC peaks[70]. Before the on-line ID calculation, $^{78}$Se and $^{80}$Se intensities were corrected for respectively, $^{1}$H$^{77}$Se and $^{1}$H$^{79}$Br interferences and the resulting $^{80}$Se/$^{78}$Se ratio was corrected for mass bias[72], as described in Supplementary Method 3 and Supplementary Fig. 21. The deconvolution of the Se mass flow chromatograms was achieved using the peak analyzer function (Fit peak-pro) of Origin2018 software following the procedure reported in Laborda et al.[33]. The semi-quantitative data for As and S speciation were obtained by deconvolution of the As and S intensity chromatograms as done for the Se mass flow chromatograms. The $r^2$ of the peak-fit were ≥0.90 for all deconvoluted chromatograms and in most cases ≥0.96. During the final analysis of all Kohala soil extracts, the water and NaOH extracts of the six Kohala topsoils (0–10 cm) were injected in triplicate to evaluate the reproducibility of the SEC-UV-ICP-MS/MS analysis, including the Se quantification by on-line ID and subsequent peak deconvolution (Supplementary Discussion 6).

## Se speciation in soil extracts by AEC-ICP-MS/MS

To compare the SEC results with those from the commonly used anion exchange chromatography (AEC), SeO$_4^{2-}$(VI), SeO$_3^{2-}$(IV) and small organo-Se species (SeMet and SeCys$_2$) were quantified in Kohala soil extracts by AEC-ICP-MS/MS using the same system than for the SEC-ICP-MS/MS analyses. The AEC separation was as described in Tolu et al.[13], i.e., column, Hamilton PRP-X100 (4.1 × 250 mm, 10 μm; Phenomenex); mobile phase, 5 mmol L$^{-1}$ ammonium citrate (ACS, ≥ 98%; Sigma-Aldrich) at pH 5.4 (fixed with 25% ammonia, Emsure ACS; Merck) with 2% methanol (Analytical grade; Merck); flow rate, 1 mL min$^{-1}$; and injection volume, 100 μL. ICP-MS/MS parameters and tuning were as described for the analysis of digests, and Se ($m/z$ 78->78; acquisition time, 0.1 ms) was analyzed with 5 mL min$^{-1}$ H$_2$ in the C/RC. Quantification was done by external species-specific calibration (i.e., Se standards prepared in the extraction matrices), and data treatment was performed using MassHunter (Agilent) software. Aqueous standards of SeO$_3^{2-}$(IV) (99.999% in 2% HNO$_3$) and SeO$_4^{2-}$(VI) (99.99% in 0.1% HNO$_3$) were obtained from Spectracer (UK, Ltd). D,L-SeMet and D,L-SeCys$_2$ standards were obtained from Sigma-Aldrich (≥99%), and stock solutions containing 1000 mg(Se) L$^{-1}$

were prepared in ultrapure water and 0.2% HCl (Suprapur, Roth), respectively. All Se species standards were stored in the dark at 4 °C.

## Calculation of enrichments factors with respect to parent rock

The enrichment factors of Se, As, S and other elements in Kohala soils were calculated using niobium (Nb), an immobile element in Kohala soils, following the Eq. (1)[41].

$$EF_i = \frac{\frac{[i]_{soil}}{[Nb]_{soil}}}{\frac{[i]_{parent\ rock}}{[Nb]_{parent\ rock}}} \times 100 \tag{1}$$

With: $EF_i$ representing the enrichment factor of element i (i.e., Se, As, S, etc) in Kohala soils, $[i]_{soil}$ corresponding to the concentration of element i in Kohala soils; $[i]_{parent\ rock}$ corresponding to the concentration of element i in Kohala parental rock; $[Nb]_{soil}$ corresponding to the concentration of niobium (Nb) in Kohala soils; and $[Nb]_{parent\ rock}$ corresponding to the concentration of Nb in Kohala parental rock. Note that the enrichment factors of NaOH-extractable Se, As, and S species with respect to parent rock were calculated with Eq. (1) by replacing $[i]_{soil}$ by the concentrations of the NaOH-extractable Se, As, or S species.

## Statistical analyses and standard deviation calculation

Bi-variate correlation coefficients were determined as 2-tailed Spearman correlation coefficients using SPSS Statistics23 (IBM) software. Box-and-whisker plots were made using the software Origin 2018.

## Data availability

All data generated in this study can be found on ETH Zurich Research Collection repository (https://doi.org/10.3929/ethz-b-000574843).

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

## Acknowledgements

We thank Eawag, ETH Zurich, and the Swiss National Science Foundation (SNF project number PP00P2_163747 for S.B., and SNF project number 200021_162422 for J.H.) for funding this work. We thank also Numa Pfenninger and Johanne LeBrun-Thauront for assistance in some laboratory work.

## Author contributions

J.T., S.B. and L.H.E.W conceptualized the study. J.T. organized, coordinated, and carried out all the lab work with help from O.H. or S.C. J.H., F.T., E.F. and O.A.C. collected the soils, parent rock and plant samples and J.H. analyzed ancillary soil parameters. J.T. conducted all data treatment and statistical analyses. J.T., S.B. and L.H.E.W. wrote the manuscript with inputs from J.H., F.T., E.F. and O.A.C, and J.T. revised the manuscripts with inputs from S.B., L.H.E.W., J.H., F.T., E.F. and O.A.C.

## Competing interests

The authors declare no competing interests.
