## [Peer Review File · Nature Communications]

Understanding soil selenium accumulation and bioavailability through size resolved and elemental characterization of soil extractsREVIEWER COMMENTS

Reviewer #1 (Remarks to the Author):

Size-resolved multi-elemental characterization of soil extracts for an improved understanding of soil selenium cycling and bioavailability

the manuscript by Tolu and coworkers provides comprehensive assessment of Se cycling in soil. The technical approach used is based on size exclusion chromatography (SEC) combined with ICP MS detection to probe metal association with various macromolecules and inorganic particles. The use of SEC for such speciation application is not novel. There are many reports in the literature for Se and other metals, especially in biological settings. Significantly fewer in environmental applications. In general the limitations of the technology (low resolution) are known however not heavily discussed here.

I find the technical content, measurement and data quality very high. The overall conclusions however are not entirely unexpected.

few technical issues:

-why arsenic is used for benchmarking? very different chemistry

-the paper in my view, especially for the chosen medium, is too data centric. I would suggest moving a large(er) portion of the data the supplementary file and focus more on the hypothesis and interpretation

Reviewer #2 (Remarks to the Author):

The authors reported a newly developed analytical method for selenium speciation in soil. The research was well designed and executed. The manuscript was well written. The reported method and related research findings are of interest particularly to readers in the field of selenium biogeochemistry. I recommend it for publication after minor revision.

Specific comments for authors' consideration:

L. 4: "selenium cycling" – this study did not address the processes of selenium biogeochemical cycling. It could be revised to selenium bioavailability or selenium chemical behaviors.

L. 8: “climate gradient” – this might not be suitable for the Big Island in Hawaii. It should be revised to “the rainfall gradient” (see page 5, line 79).

L. 11: All the results presented in the abstract are only applicable to the Kohala soils. In addition, the method limitation should also be included (such as those mentioned on page 7, line 138-139).

L. 27: delete (VI) and (IV).

L. 29: bulk Se (0). Nanoscale elemental Se can be partially available to plants.

L. 31-32: This is overstated. Numerous studies have been done using different methods or experimental approaches, and only some of research findings are controversial.

L. 34: Selenium organic compounds can be transformed to Se oxyanions through oxidation, while Se-containing/adsorbing organic matter can release Se back to soil solution through degradation of non-humic substances or soil organic matter.

L. 48: “questionable” – What are particularly questionable? the extraction solutions or the extraction procedures? What are the differences between those used in other studies and ones (water and NaOH solution) used in this study?

L. 69: In NaOH extracts,

L. 74: Not clear. “its accumulation” in soil or plant or both?

L. 85: “anoxic conditions and intensive leaching” – In general, intensive leaching means good drainage and less standing water flooding or less anoxic. But in this study both occur at Site 6. What special at Site 6, with heavy surface run-off?

L. 111: See comments above (L. 48).

L. 114: for Size separation, 0.3-100 kDa was studied (and thoroughly optimized). Will this size range reflect the particle composition in soils?

L. 138-139: Is this the limitation of this newly developed method? If so, then this shall be highlighted in this method paper.

L. 140: “large Se(0) nanoparticles (50-500 nm)” – Did you mean small nanoparticles of <50 nm? What is the justification for this classification when nanoparticles are commonly defined as particles of <100 nm in diameter?

L. 149: soil horizon”s” A and B. What was the depth of each horizon?

L. 150: which “were”

L. 166-167: small “amounts of” hydrophilic organic Se?

L. 176: How to define “older SOM”? Also, “higher SOM degradation rates” will need adequate soil water content for soil microbes.

L. 186: Microbe-produced (nano)particles commonly have a size of >100 nm.

L. 223: What are the common particle size range for those Fe (oxy)hydroxides?

- L. 226: Did you mean “As for S”?
- L. 256-257: change it to “airborne S”?
- L. 264: Change it to “soil development processes”
- L. 267: delete “study”
- L. 271: “less important ...” It should be “less net input” from the atmosphere after volatilization.
- L. 340: “organic Se”? or OM-bond Se? or Se-containing soil organic matter?
- L. 352: change it to “organic-trace element complexes”
- L. 356: predict crop “Se” contents?
- L. 398: using a “tuning” solution?
- L. 477: as reported in 80?
- L. 495: prepared in, respectively,

Reviewer #3 (Remarks to the Author):

Overall this paper describes a thorough and carefully tested approach to determination of Se speciation in alkaline and water extracts. The method is applied to a series of soils covering a range of Se, OM and Fe contents which forms the bulk of the discussion in the manuscript. The development of the method is largely confined to supplementary information. Unfortunately this results in the continual quoting of numerical values in the text that does not enhance the clarity or readability of the manuscript. Please consider incorporating key tables into the main manuscript rather than reproducing individual numbers in the text and referring the reader to the SI for further details.

There are a number of typographical corrections and clarifications that should also be made to the manuscript;

Lines 96-102: Please clarify which horizon(s) is/are being discussed. At line 100 it is stated that Se concentrations decrease S5-S6' yet this is only the case for Horizon A not B which appears to increase although it is not clear if this is statistically significant?

Line 67: hydroxide not hydroxides

Table S2: Fecrystalline not 'Fecristalline' in column 11.

Figure 1: Please state what the errors bars represent SD or SEM? Not all are visible – please confirm if this is because they are within the symbol? Difficult to distinguish the colours used for S5 and S6 also the same colours are not applied to Horizon B?

Line 127 & Figure 3: I question the validity of Figure 3 and comparison with AEC-ICP-MS. AEC-ICP-MS approaches do not detect Org-Se species. I suggest removing this Figure. Also in addition to quoting average recoveries please provide the range for both extracts.

Line 193: Figure b – should this be 5b?

Line 201: 'allowing estimation of' not allowing to estimate'

Line 208: not only 'are' the association of As and Se to SOM different. Please amend the position of 'are'

Line 213: 'associated with'' not 'associated to' (twice)

Line 215: please remove 'the' from before 'so-called'

Line 219: 'has' not 'had'

Line 225: 'differs' not 'differ'

Line 278: non-extracted? Not extracted? Unclear.

Line 290: 'more' or 'greater' not 'stronger'?

Line 310: 'very close to' not 'very close from'

Line 311: 'Aside from'

Line 346: collections

Line 348: simplify to 'can be combined with' removing 'directly hooked or'

Line 462-476: Please clarify the isotope used as the post UV-detector spike. In Section 2.1.1 of the SI, Fig S4, Table S3 it is given as ⁷⁷Se whereas here the isotope discussed is ⁷⁸Se.

Line 470: consisted 'of' not 'in'.

Line 492: 'Aqueous standards' not 'liquid standards'

Supplementary Information

Table S2: 'Fecrystalline' not 'Fecristalline' in column 11.

Section 2.1.1: Discussion of the choice of mobile phase is confusing. The justification for not using ammonium citrate is to reduce C content but you are adding MeOH (to enhance Se sensitivity I assume) which will increase C content. The discussion then continues to discuss ammonium phosphate as a mobile phase. Please clarify which eluents were tested and at what concentrations. Please also provide information on the extent of sensitivity loss during long runs.

Figure S3: Please clarify what the numbers given on the lines represent. Please check the r^2 value for polystyrene sulfonate in panel A – it appears to be an r^2 for a line through some of the data points? Also for Pullulan standards in Panel C.

Section 2.1.5: Reference is made to phosphate extracts – please clarify. Only NaOH and water extracts are referred to elsewhere.

Overall the technical aspects of the manuscript are quite challenging to follow because they are referred to in the results and discussion, separate materials and methods section which follows the discussion, and also in the SI where the testing and development of the method is described. To assist the reader I suggest inclusion of a table in the SI that details all the parameters tested to develop the method.

Response to reviewer's comments

First, we would like to thank the three reviewers for their useful comments, which greatly helped improving the clarity of the paper. Below are the point-by-point responses to all comments (in blue). In these responses, the text that was corrected or added in the manuscript is indicated in italics. In the manuscript and supplementary information, the changes are shown using the track changes feature of word processor.

Reviewer #1 (Remarks to the Author):

The manuscript by Tolu and coworkers provides comprehensive assessment of Se cycling in soil. The technical approach used is based on size exclusion chromatography (SEC) combined with ICP MS detection to probe metal association with various macromolecules and inorganic particles. The use of SEC for such speciation application is not novel. There are many reports in the literature for Se and other metals, especially in biological settings. Significantly fewer in environmental applications.

We first would like to thank the reviewer for acknowledging the comprehensiveness of the dataset presented in our manuscript. Regarding the novelty of the approach, SEC combined with ICPMS detection has indeed been previously used to probe metal association with organic macromolecules and, in a lesser extent, with inorganic particles, in biological samples and few types of environmental samples (i.e., model systems, compost leachates, and freshwaters). However, as the reviewer pointed out, SEC-ICP-MS is still scarcely used in environmental sciences due to the complexity of the system (samples matrices and low trace elements contents), and our study is the first developing and using a SEC-UV-ICP-MS method for a broad characterization of the chemical forms of Se as well as S and As in soils.

More precisely, our method development and its application to Kohala soils is novel for the following reasons:

- 1) **SEC-ICP-MS has never been applied to determine Se speciation in extracts of natural soils and, more generally, in environmental samples, where Se is present at trace level ($\mu\text{g L}^{-1}$).** Previous studies on Se using SEC-ICP-MS applied this technology to biological samples rich in Se (i.e., extracts from yeast or Se-accumulator plants containing $>1 \text{ mg(Se) L}^{-1}$) (ref. 36 and 37 in revised manuscript) or to a model system containing reference OM materials, Fe(III) and Se(IV) at a concentration of 1 mg(Se) L^{-1} (ref. 24 in revised manuscript). In contrast, the complex soil extracts analyzed in our study only contained $0.8\text{-}80 \mu\text{g(Se) L}^{-1}$ (cf. new Table 1 in revised manuscript). To better inform on previous studies that used SEC-ICP-MS and clarify the novelty of our work (with respect to previous studies), we have added the following sentence in the introduction: “*SEC-UV-ICP-MS has previously been used to probe metal associations with organic macromolecules and inorganic particles in model systems²⁴, compost leachates³³, freshwaters^{34,35} and biological samples^{36,37}, and we optimized it for 0.1 M NaOH and ultrapure water extracts of natural soils.*” (L. 58-61 in revised manuscript).
- 2) **As pointed out by the two other reviewers, we thoroughly optimized the SEC separation to analyze Se speciation in soil extracts that represent complex matrices** as indicated in the manuscript (L. 60-61, L. 111-113, and L. 117-122 in revised manuscript) and described in details in the supplementary information (Section 3 in revised Supplementary Information, p. 6-13). Indeed, we tested different types of SEC columns and mobile phases to ensure optimal resolution of the trace elements separation as well as optimal Se species recovery. In contrast, most previous studies did not optimize the SEC separation with respect to resolution of the element separation and recovery of the analyzed elements.
- 3) **Our work is the first quantifying Se species in environmental samples by on-line isotope dilution LC-ICP-MS** (indicated L. 112-113). Performing on-line isotope dilution with Se is challenging given the numerous interferences existing for Se with ICP-MS detection. Our paper thus provides, in addition

to new SEC separation conditions, the required parameters and interferences corrections to perform on-line ID LC-ICP-MS analysis of Se speciation in complex environmental samples.

- 4) In our work, the SEC separation was optimized to detect organic matter not only using an UV detector, as conventionally done, but also using the ICP-MS/MS (cf. Section 3.1, p.6-8 and section 3.2.3, p. 12-13 in revised supplementary information). While an UV detection only detects chromophoric organic matter, the ICP-MS/MS provides information on the SEC elution of total carbon (cf. Figure 2 in revised manuscript). **Our simultaneous detection of total C, UV and trace elements demonstrates the presence of a non-aromatic, aliphatic organic fraction in water extracts** (cf. Section “Characterization of Se species in soil extracts with SEC-UV-ICP-MS/MS” in revised manuscript and section 3.3.3 in revised supplementary information, p. 16-18) **to which S, but not Se and As, is associated with, and thus provides new information on the types of OM present in soil water extracts and on the speciation of water-soluble Se, As and S.**

In general the limitations of the technology (low resolution) are known however not heavily discussed here. Concerning the low resolution of the size exclusion chromatography, it is indeed known that SEC has low resolution compared to other types of liquid chromatography such as anion exchange (AEC) or mixed mode (MMC) chromatography. However, these latter separation techniques do not elute all Se (and other trace elements) species. As we explain(ed) in the manuscript, a large fraction of Se species (even up to 100% for some soils) remain unrecovered in soil extracts with such LC separation; we wrote at L. 50-54 in revised manuscript “*Furthermore, speciation analysis of soil extracts using selective reduction of Se(IV)^{19,20,22,23} or chromatographic separation of Se oxyanions and amino acids was sometimes performed^{13,14,31,32}, but the former only informs on Se oxidation states whereas a large share of extracted Se species (20-100%) remains unrecovered by commonly used liquid chromatographic methods.*”, and at L. 133-135 “*In contrast, 43-100% of Se_{water} and 38-90% of Se_{NaOH} are unrecovered with the conventional AEC-ICP-MS/MS method, with which only Se oxyanions were detected (Table 1).*”

In addition, although SEC-UV-ICP-MS technology is limited in resolution, it currently provides the highest resolution on Se speciation in natural soils (not contaminated/Se-spiked soils) and other environmental samples (e.g., sediment, freshwaters). As explained in the introduction (L. 44-47) in revised manuscript, solid-phase techniques have too high detection limits to be applied on environmental samples that are not substantially spiked or enriched with Se. Of course, an important next step to our work is to push the resolution of the Se speciation analysis by SEC-UV-ICP-MS/MS forward by combining our SEC method to high-resolution mass spectrometry to identify the organic Se compounds at the molecular level. This is described in the section “*Outlook*” of our manuscript: “*Furthermore, our SEC method can be combined with high-resolution mass spectrometry after appropriate sample or fraction pre-concentration. This will certainly allow structural elucidation of organic forms of trace elements and thus better inform on the mechanisms for trace element binding or incorporation into SOM as well as on the degradation and plant availability of the organic-trace element complexes or compounds*” (L. 350-354 in revised manuscript).

I find the technical content, measurement and data quality very high. The overall conclusions however are not entirely unexpected.

We would like to thank the reviewer for the positive feedback on the technical content, measurement and data quality. Regarding the comment that the overall conclusions “are not entirely unexpected”, we clearly mentioned in our introduction that previous studies and reviews speculated on specific processes and forms of Se in soils, and more precisely that:

- A large quantity of soil Se is organic Se.
- Se accumulation in soils is strongly controlled by its association with soil organic matter, except in mineral soils such as volcanic soils.

- Organic Se is of low plant-availability or is an important pool storing Se in soils and releasing plant-available Se after organic Se degradation or Se desorption from organic matter.

However, none of these hypotheses could be clearly proven. These hypotheses were based on 1) indirect evidence, such as relationships between soil Se or plant Se concentrations and soil organic matter concentrations; 2) high concentrations of Se found in extracts assumed to extract specifically for organic Se (e.g., NaOH, TMAH, NaOCl); and 3) on assumed similarity between Se and S or As. Besides providing clear insights on these hypotheses for natural soils (not contaminated/Se-spiked soils), we could further differentiate “organic Se” by identifying multiple organic fractions and associations with other trace elements.

Few technical issues:

-why arsenic is used for benchmarking? very different chemistry

Although in more biologically oriented research fields arsenic and selenium are known to have very different (bio)chemistry, it is common in environmental research that arsenic and selenium are compared: both elements are classified as metalloids and exist as oxyanions in soils and aquatic systems. Moreover, both Se(IV) and As(V) oxyanions are known to have a strong affinity for sorption to Fe (oxy)hydroxides and with respect to association with natural organic matter, it has been hypothesized that Se oxyanions, similarly to As oxyanions, are complexed to Fe(III) which is complexed to organic matter (ternary complexation). This hypothesis is presented in the introduction of the revised manuscript at L. 37-43 “*Nevertheless, hypotheses about the nature of organic Se were made based on its chemical similarity to sulfur (S) and arsenic (As)^{11,12,16,24}. Similarly to S²⁵, organic Se may consist of Se proteins and metabolites and/or Se covalently bound to soil organic matter (SOM) after incorporation of microbially produced hydrogen selenide(-II) into SOM. Similarly to As^{26,27}, organic Se may consist of oxyanions bound to SOM (through ternary complexation) and/or adsorbed onto mineral (nano)particles coated by SOM. However, the combined speciation of soil Se, S and As has not been investigated previously.*” We then concluded on the nature of organic Se based on our SEC data for both Se and As in the results and discussion section at L. 226-228 in revised manuscript “*By showing that up to 11% of Se_{NaOH} is associated with the Fe and As enriched OM fraction, our data suggests that Se ternary complexation may occur in soils, however, is not a dominant mechanism of Se association to SOM.*”

-the paper in my view, especially for the chosen medium, is too data centric. I would suggest moving a large(er) portion of the data the supplementary file and focus more on the hypothesis and interpretation

We agree with the reviewer that a significant amount of data is presented in the paper, but this is a key requirement for this new broad characterization of Se speciation and behavior in this complex natural system. Considering that we already placed many details on the method development and environmental results in the Supplementary Information, and that we present the data in a comprehensive and condensed way in the main manuscript (important for the reader to understand our results and conclusions), we think that we should not put further information in the supporting files. In addition, the 3rd reviewer suggested to move data (in the form of tables) from the supplementary files into the manuscript itself.

Moreover, we (had) introduced our working hypotheses/questions in the introduction (cf. L. 30-43 in revised manuscript) and interpreted our data to conclude on these working hypotheses/questions in the results and discussion part of the manuscript (cf. sections “*Organic Se is a dominant form in Kohala soils*”, L. 192-198; “*Extractability, size distribution and speciation largely differ between Se and As*”, L. 226-232; “*Se and S extractability are similar but their speciation strongly differs*”, L. 248-266; “*Soil enrichment of Se, As and S explained by their speciation*”, L. 301-303 and “*Relationships between Se and S concentrations in plants and their water-soluble speciation*”, L. 327-330 and L. 337-340 in revised manuscript).

Reviewer #2 (Remarks to the Author):

The authors reported a newly developed analytical method for selenium speciation in soil. The research was well designed and executed. The manuscript was well written. The reported method and related research findings are of interest particularly to readers in the field of selenium biogeochemistry. I recommend it for publication after minor revision.

We would like to thank the reviewer for this positive feedback on the quality of the manuscript and interest for the scientific community as well as for the detailed review.

Specific comments for authors' consideration:

L. 4: “selenium cycling” – this study did not address the processes of selenium biogeochemical cycling. It could be revised to selenium bioavailability or selenium chemical behaviors.

We agree, and replaced “*selenium cycling*” by “*selenium accumulation in soils and plant-availability*” (L. 4 in revised manuscript).

L. 8: “climate gradient” – this might not be suitable for the Big Island in Hawaii. It should be revised to “the rainfall gradient” (see page 5, line 79).

We agree, and thus replaced “*climate gradient*” by “*rainfall gradient*” (L. 8 in revised manuscript).

L. 11: All the results presented in the abstract are only applicable to the Kohala soils. In addition, the method limitation should also be included (such as those mentioned on page 7, line 138-139).

We agree, and clarified that all the results presented in the abstract are from the Kohala rainfall gradient by changing the sentence as follows: “*Combined soil speciation data with concentrations in parent rock and plants along the Kohala rainfall gradient suggest that selenium association with organic matter limits its mobility and availability for plants*” (L. 10-12 in revised manuscript). Analyzing more soils with the new SEC-UV-ICP-MS/MS method is indeed important to confirm the role of organic matter on soil Se cycling suggested by our broad characterization of soil Se speciation in Kohala soils, as proposed in the “*Outlook*” (L. 354-356).

Concerning the comment “the method limitation should also be included (such as those mentioned on page 7, line 138-139)”, we agree that the term “*complete*” within the sentence “...*providing a complete characterization of selenium in soil extracts*” is not ideal given that up to 45% of Se in some water extracts do not elute out of the optimized SEC-UV-ICP-MS/MS method. We therefore replaced “*complete*” by “*advanced*” (L. 6 in revised manuscript), considering that the abstract in *Nature Communication* should be of 150 words and should not include technical details.

L. 27: delete (VI) and (IV).

To make clear that these numbers indicate the oxidation states of selenate and selenite, which is an important information to provide to the readers, we rewrote the sentence as follows: “*Various inorganic Se species exist in soils, i.e., selenate (SeO_4^{2-} ; oxidation state, +VI) and selenite (SeO_3^{2-} ; +IV) present...*” (L. 25-26 in revised manuscript).

L. 29: bulk Se (0). Nanoscale elemental Se can be partially available to plants.

We agree, and therefore rewrote the sentence as follows: “...*whereas Se(0) (nano)particles, metal selenides and Se oxyanions co-precipitated with minerals are much less, or not at all, mobile and plant available*” (L. 27-30 in revised manuscript).

L. 31-32: This is overstated. Numerous studies have been done using different methods or experimental approaches, and only some of research findings are controversial.

We agree, and thus rewrote the sentence as follows: “*Organic Se species are also present in soils¹²⁻¹⁵ but they are only indirectly characterized (by “selective” extractions or difference between total and inorganic Se) and their behavior remain unclear in different aspects¹⁶.*” (L. 30-32 in revised manuscript).

L. 34: Selenium organic compounds can be transformed to Se oxyanions through oxidation, while Se-containing/adsorbing organic matter can release Se back to soil solution through degradation of non-humic substances or soil organic matter.

We agree and have thus complemented the description of the previously proposed hypotheses on organic Se plant-availability as follows: “*...others proposed it as an important plant-available Se pool through its direct uptake (e.g., seleno-amino acids), its oxidation into plant-available Se oxyanions, and/or the release of complexed Se oxyanions into solution during SOM degradation²⁰⁻²³.*” (L. 32-35 in revised manuscript).

L. 48: “questionable” – What are particularly questionable? the extraction solutions or the extraction procedures? What are the differences between those used in other studies and ones (water and NaOH solution) used in this study?

We think that the lack in selectivity of these previously used extraction procedures mainly comes from the used extraction solutions, although the extraction conditions can also lead to incomplete extractions of targeted Se species, contamination issues, or Se species transformation. To provide the readers with meaningful information on what is known (and unknown) about the selectivity of selective extraction procedures, we have added a text in the supplementary information (cf. Section 1.2 “*Selectivity of “selective” extractions*” in revised supplementary information; p. 2-3). Reference to this supplementary text is done in the main manuscript as follows: “*However, selectivity of these extractions is questionable²⁸⁻³⁰ (Supplementary Section 1.2).*” (L. 49-50 in revised manuscript).

Regarding the differences between the extraction solutions we used in our study and those used in other studies, there are a large number of extraction solutions employed to determine water-soluble and organic Se, and in our study, we selected the ones that are the most widely used. To clarify this point, we have:

- added in the introduction, the sentence “*NaOH is widely used to estimate organic Se quantity and characterize SOM, while ultrapure water is employed to assess mobile and plant-available Se^{11,16,38} (Supplementary Section 1.1).*” (L. 61-63 in revised manuscript).

- added in the supplementary information, an overview of previously used extraction solutions (cf. Section 1.1 “*Commonly used “selective” extraction*” in revised supplementary information, p. 1) and a comparison between them in term of Se extraction efficiencies (cf. Section 1.3 “*Comparison between commonly used “selective” extractions of water-soluble and organic Se*” in revised supplementary information, p. 2-3).

L. 69: In NaOH extracts,

We agree, and replaced “*alkaline*” by “*NaOH*” (L. 74 in revised manuscript).

L. 74: Not clear. “its accumulation” in soil or plant or both?

We clarified this sentence as follows: “*...the dominant organic nature of Se in soils is pivotal to understand its accumulation in soils and its low plant-availability*” (L. 78 in revised manuscript).

L. 85: “anoxic conditions and intensive leaching” – In general, intensive leaching means good drainage and less standing water flooding or less anoxic. But in this study both occur at Site 6. What special at Site 6, with heavy surface run-off?

We understand that the co-occurrence of “anoxic conditions and intensive leaching” seems contradictory but anoxic conditions can develop during events of heavy rainfall, where rainfall may bring in more water that can

be leached out, leading to water saturation in the soil. To clarify this point, we revised the corresponding sentence as follows: “*At humid sites S5-S6 (2163-3123 mm y⁻¹)³⁹, mean annual precipitation is well above potential evapotranspiration resulting, at site S6, in intensive leaching and periods of anoxic conditions during heavy rainfalls where rainfall rates outpace leaching rates leading to water saturation^{40,41,43}.*” (L. 89-92 in revised manuscript). More precisely, leaching intensity along the Kohala rainfall gradient has first been estimated by Chadwick et al. (2003; reference 40 in revised manuscript) using the ratio between the available pore volume in the top meter and the average annual depth of water penetration. They showed that the Kohala soils that receive rainfall >3000 mm y⁻¹ (which is the case for site S6) have experienced nearly seven pore volumes of leaching water exchange annually. Additionally, important net loss of different elements (e.g., base cations –i.e., Ca, Na, and K-, Si, Al, Fe, and P) at these wettest Kohala sites were reported in this paper as well as in a few others (e.g., reference 41 in revised manuscript). Concerning the more reducing conditions in soils at Kohala site S6 (rainfall 3123 mm y⁻¹), we observed in our study, a decrease in the concentrations of Fe_{crystalline}, Fe_{amorphous} (Figure 1 in revised manuscript) and in the enrichment factor of Fe between sites S5 and S6 that coincide with decline in soil As concentrations and enrichment factor with respect to parent rock (Figures 1 and 5 in revised manuscript). Altogether, this potentially indicates increased mobility of Fe and As due to reductive dissolution of (oxy)hydroxides at site S6. In line with this interpretation, Schuur et al. (2001; reference 43 in revised manuscript) reported a decrease in the soil redox potential with increasing rainfall amount from 2000 to 3200 mm y⁻¹ along the Maui (Hawaii) rainfall gradient where soil developed on lava flows from the Kula volcanic series (mean age 410 000 yr). Vitousek and Chadwick (2013; reference 41 in revised manuscript) also reported anoxic conditions and Fe mobility (probably due to reductive dissolution of (oxy)hydroxides) at sites with rainfall >2500 mm y⁻¹ of the Kauai (Hawaii) rainfall gradient.

L. 111: See comments above (L. 48).

Please see our answer above.

L. 114: for Size separation, 0.3-100 kDa was studied (and thoroughly optimized). Will this size range reflect the particle composition in soils?

Our main objective was to quantify and characterize the organic forms of Se (together with those of S and As) in soils after extraction with water and NaOH. As demonstrated in the manuscript, the selected column and conditions are the most efficient at separating different organic Se, As and S fractions from Se, As and S free oxyanions and Se, As, and S associated with small (organo)mineral nanoparticles (<20-40 nm) in soil extracts, which is a critical point to achieve our objective.

Methods used to estimate (e.g., sequential extractions and total element quantification or hydride generation-AFS) or to determine (e.g., LC-ICP-MS) soil Se speciation never reflect the particle composition of soils because of soil sample preparation (e.g., soil sieving) and the extraction step (followed by centrifugation and filtration of the soil extracts at max. 0.45 µm, needed before injection into analytical instruments). Soils include sand (0.05-2 mm), silt (2-50 µm) and clay (<2µm) size fraction, and even centimeter scale particles; these latter being generally removed before trace element analysis by soil sieving (as done in our study) because trace elements are well-known to be enriched in the clay and silt fractions. The common filtration of the extracts at 0.45 µm and the size separation of our optimized SEC (0.3-100 kDa and <~40 nm, which is the column pore size) do also not entirely reflect the soil extract particle composition. Indeed, the few previous studies investigating this reported that water extracts can contain particles of size up to few µm (references 61 and 62 in revised supplementary information) and that alkaline extracts contain particles of size between 10 and 150 nm (references 63 and 64 in revised supplementary information). However, although the size separation of SEC methods cannot cover the complete particle size range of the soil extracts, our optimized SEC method recovers all Se (and S) present in NaOH extracts, i.e., the SEC Se species recoveries were 95-112 (101±5) % of total Se in NaOH extracts (n=25 soils; cf. new added Table 1 in revised manuscript). Therefore, there is no need for a method with a wider size range to determine Se speciation in alkaline extracts. In water extracts, we could recover 58-109 (81±15) % of

total Se in water extracts (n=25 soils; cf. Table 1 in revised manuscript), and by quantifying elements in water extracts after filtration at 20 nm, we could observe that the unrecovered Se species by SEC are too big to elute out of the column (L. 141-148 and new added Table 2 in revised manuscript). Therefore, a SEC column specific for larger nanoparticles or field flow fractionation is needed to identify and study the unrecovered fraction of Se in water extracts. We incorporated these further clarifications including the cited papers in revised manuscript or revised supplementary information, as described in details in the answer to next comment.

L. 138-139: Is this the limitation of this newly developed method? If so, then this shall be highlighted in this method paper.

The fact that some Se in Kohala soil water extracts (up to 45%) does not elute out of our SEC-UV-ICP-MS/MS method may be seen as a limitation of this newly developed method. However, we would like to point out here that the multi-elemental SEC-UV-ICP-MS/MS data (i.e., including SEC recoveries of elements such as S, Cu, Fe, As) combined with multi-element quantification of water extracts filtered at 20 nm clearly indicates for the first time the presence of Se associated to (organo)mineral nanoparticles of size >20 nm in soil water extracts. To better communicate on this aspect, we made the following changes:

1) We changed the title of the results and discussion section presenting the method, which is now as follows: “*Characterization of Se species in soil extracts with SEC-UV-ICP-MS/MS*” (L. 110 in revised manuscript).

2) We have added, in this section, the sentence “*The fraction of Se_{water} that was not recovered by our SEC method (<45%) is thus likely associated with >20-40 nm (nano)particles, and future studies are needed to better characterize this Se fraction in soil water extracts using a SEC column specific for (nano)particles⁴⁶ or field flow fractionation⁴⁵.*” (L. 145-148 in revised manuscript).

3) We specified at the end of this section that our method identifies and quantifies “*organic Se of different size and chemical properties together with free Se oxyanions and Se associated with small (organo)mineral nanoparticles (<20-40 nm)*” (L. 152-155 in revised manuscript).

4) We have added a discussion on how the optimized SEC method reflects the particle composition of soils and soil extracts (as answered to previous comment) in revised supplementary information (cf. Section 3.3.1 “*Targeted size and size determination with the optimized SEC-UV-ICP-MS/MS method*”, p- 13-14).

5) We have added, in the “*Outlook*”, the sentence “*Using SEC columns specific for nanoparticles⁴⁶ or field flow fractionation⁴⁵ would also be important to characterize the size fractions >20-40 nm of water-soluble Se and As.*” (L. 362-363 in revised manuscript).

L. 140: “large Se(0) nanoparticles (50-500 nm)” – Did you mean small nanoparticles of <50 nm? What is the justification for this classification when nanoparticles are commonly defined as particles of <100 nm in diameter? We actually meant “50-500 nm” and not “<50 nm” as to the best of our knowledge, there is no study showing the formation of Se nanoparticles in size <50 nm in contaminated soils or bacterial cultures. We corrected the sentence as follows: “*...although only Se(0) nanoparticles of 50-500 nm size⁵⁰⁻⁵² have been found in contaminated soils or bacterial cultures...*” (L. 148-149 in revised manuscript). We also checked the use of the term “nanoparticles” everywhere in the manuscript.

L. 149: soil horizon”s” A and B. What was the depth of each horizon?

The depth of each soil horizon is different for each of the six study sites as can be seen in Table S1. To clarify where to find the information on soil horizon depths, we have added:

- the sentence “*The soil samples cover A- and B-horizons, with the horizon depths being different at each site (Table S1).*” in the first section of the Results and Discussion part (L. 84-86 in revised manuscript)

- the sentence “*Correspondence between soil horizon and soil depth is given in Table S1 in Supplementary Section 2*” in revised captions of Figures 1 and 3-5.

L. 150: which “were”

Corrected (L. 160 in revised manuscript).

L. 166-167: small “amounts of” hydrophilic organic Se?

We do not refer to a small amount of hydrophilic organic Se, but we are discussing the identity of Se compounds found in the small size fraction of organic Se. We rewrote the sentence as follows: “*Potentially, the small hydrophilic organic Se fraction could include Se metabolites such as selenosugars...*” (L. 177-180 in revised manuscript).

L. 176: How to define “older SOM”? Also, “higher SOM degradation rates” will need adequate soil water content for soil microbes.

Our use of “older SOM” was indeed vague, and there is no carbon dating data available to support the use of the term “older SOM”. There is also no SOM degradation rate calculated for studied Kohala soils. Indeed, the paper we are referring to in this sentence investigated soils from another rainfall gradient in Hawaii. Still, there is water available in the Kohala driest sites as there is also growing vegetation at these sites (i.e., *Buffer Grass* that we analyzed in our study), although in a much lower extent as in the (sub)humid sites. Because referring to “older SOM” and “higher degradation rates” is not central to our interpretation of the difference in Se distributions among the size and chemical organic fractions between the driest and (sub)humid sites, and to provide a thorough discussion here as well, we removed reference to older SOM and higher SOM degradation rates in the referred sentence. The sentence reads now as follows: “*At the driest sites (S1-2), the vegetation cover is scarce, which result in lower inputs of fresh OM than in (sub-)humid sites S3-6^{40,41,43}.*” (L. 185-186 in revised manuscript).

L. 186: Microbe-produced (nano)particles commonly have a size of >100 nm.

We agree but as there are studies reporting microbially-produced Se(0) nanoparticles with a size down to 50 nm, we changed the sentence as follows: “*... and/or by favoring (a)biotic Se reduction into Se(0) nanoparticles, , which commonly have sizes >50 nm⁵⁰⁻⁵².*” (L. 196 in revised manuscript).

L. 223: What are the common particle size range for those Fe (oxy)hydroxides?

Fe(III) (oxy)hydroxides particles formed during the reaction of Fe(III) with humic substances are usually in the low nanometer range (few nm to a few tens of nm) as opposed to those formed by biotic reduction of Fe(III). There was no size characterization of the particles formed in the study of Martin et al. (2017; reference 24 in revised manuscript) but similar experiments were carried out in the past to detect and characterize Fe(III)-organic matter (OM) complexes (Karlsson and Persson, 2010; reference 59 in revised manuscript) and As-Fe-OM ternary complexes (Mikutta and Kretschmar, 2011; reference 26 in revised manuscript). In the work of Mikutta and Kretschmar (2011), the Fe(III) (oxy)hydroxides particles formed during the incubation of Fe(III) with reference humic substances are mentioned to be of nanometer sizes. We clarified this by revising the corresponding sentence as follows: “*However, the SEC-ICP-MS method used in this earlier study was not resolute enough to distinguish between these complexes and Se(IV) adsorbed to nanometer-sized Fe(III) (oxy)hydroxides that form in such model system^{26,59}.*” (L. 230-232 in revised manuscript).

L. 226: Did you mean “As for S”?

No, we meant “*Similarly to Se*” and thus modified the sentence to avoid confusion as follows: “*Similarly to Se, a large share of Kohala soil S is extracted by NaOH (53±12% versus 65±12% of soil Se; Figure S14), and 1.8±1.3% of soil S is extracted by water...*” (L. 235-237 in revised manuscript).

L. 256-257: change it to “airborne S”?

We changed “*gaseous atmospheric S*” by “*airborne S*” (L. 265 in revised manuscript).

L. 264: Change it to “soil development processes”

We replaced “*soil processes*” by “*soil development processes*” (L. 273 in revised manuscript).

L. 267: delete “study”

We have deleted “*study*” in front of “*Kohala soils*” (L. 276 in revised manuscript).

L. 271: “less important ...” It should be “less net input” from the atmosphere after volatilization.

We agree but we removed this sentence discussing atmospheric As versus Se inputs to Kohala soils (which was not a focus of our study) to fulfill the journal’s word number requirement (L. 279 in revised manuscript).

L. 340: “organic Se”? or OM-bound Se? or Se-containing soil organic matter?

Thank you for reflecting on the term to use in this sentence. We kept the term “*organic Se*” because it includes organic Se compounds that are produced by (micro)organisms, Se that is complexed to OM and Se covalently bound to OM. Also, “*organic Se*” was defined in the introduction where the different hypotheses on organic Se forms are presented (L. 30-43 in revised manuscript).

L. 352: change it to “organic-trace element complexes”

We have added the term “*complexes*” but kept the term “*compounds*” because our sentence also refers to organic trace element forms that involve covalent binding (and not complexation), which are known to exist for S and are likely to exist for Se as well (L. 354 in revised manuscript).

L. 356: predict crop “Se” contents?

Corrected (L. 357 in revised manuscript).

L. 398: using a “tuning” solution?

Corrected (L. 402 in revised manuscript).

L. 477: as reported in 80?

80 was the reference number of a study who reported parameters for deconvolution of SEC chromatogram. We therefore replaced “*as reported in 80*” by “*following the procedure reported in Laborda et al.³³*” (L. 499 in revised manuscript).

L. 495: prepared in, respectively,

Corrected (L. 521 in revised manuscript).

Reviewer #3 (Remarks to the Author):

Overall this paper describes a thorough and carefully tested approach to determination of Se speciation in alkaline and water extracts. The method is applied to a series of soils covering a range of Se, OM and Fe contents which forms the bulk of the discussion in the manuscript. The development of the method is largely confined to supplementary information. Unfortunately this results in the continual quoting of numerical values in the text that does not enhance the clarify or readability of the manuscript. Please consider incorporating key tables into the main manuscript rather than reproducing individual numbers in the text and referring the reader to the SI for further details.

We would like to thank the reviewer for the positive feedback on our manuscript and the detailed review. The description of the method development is indeed confined to the supplementary information, and we understand that the continual quoting of numerical values in the text (from tables given in the supplementary information)

created difficulty in reading the SEC-UV-ICP-MS/MS method development and performance section of the main manuscript. To increase the readability of this section (“*Characterization of Se species in soil extracts with SEC-UV-ICP-MS/MS*”), we reduced the quoting of individual numbers in the text and added two tables in the manuscript (see L. 110-155 and Tables 1-2 in revised manuscript). Along with these changes, previous Table S4 and Figure S13 in the supplementary information were removed as the data are now shown in Table 1, and Figure S11 was removed as the data are now shown in Table 2.

There are a number of typographical corrections and clarifications that should also be made to the manuscript; Thanks for pointing them out to us. All of them were addressed (see below).

Lines 96-102: Please clarify which horizon(s) is/are being discussed. At line 100 it is stated that Se concentrations decrease S5-S6’ yet this is only the case for Horizon A not B which appears to increase although it is not clear if this is statistically significant?

We clarified which horizon is being discussed within each of the sentences of the referred paragraph as follows: “*Between S5 and S6 (the wettest site), the concentrations of As in horizon A clearly decrease showing a similar trend as amorphous Fe (oxy)hydroxides, whereas S concentrations in horizon A increase similarly to SOC concentrations. Like As, Se concentrations in horizon A decrease between S5 and S6 but to a lesser extent. In the horizon B, Se and S show similar trends than SOC while As is again similar to amorphous Fe (oxy)hydroxides.*” (L. 104-108 in revised manuscript). As pointed out by the reviewer, only Se concentrations in soil horizon A clearly decrease between sites S5-S6 and we prefer to not discuss the trend in Se concentrations in horizon B, which is insignificant when considering the standard deviation values.

Line 67: hydroxide not hydroxides

Corrected (L. 72 in revised manuscript).

Table S2: Fecrystalline not ‘Fecristalline’ in column 11.

Corrected (see Table S2 in revised supplementary information, p. 5).

Figure 1: Please state what the errors bars represent SD or SEM? Not all are visible – please confirm if this is because they are within the symbol? Difficult to distinguish the colours used for S5 and S6 also the same colours are not applied to Horizon B?

We clarified that the errors bars represent standard deviation (SD) in the captions of Figure 1 as well as Figures 3-7 in revised manuscript. It is also now explicitly written in the revised Figure 1 caption that i) there are no SD available for Fe_{crystalline} and Fe_{amorphous} concentrations; and ii) when the error bars are not visible for SOC, Se, As, and S concentrations, this is because they are within the symbol. Finally, we revised Figure 1 as recommended, i.e., the color for site S5 was changed to make it easier to distinguish between sites S5 and S6, and the same colors for horizons A and B have been used.

Line 127 & Figure 3: I question the validity of Figure 3 and comparison with AEC-ICP-MS. AEC-ICP-MS approaches do not detect Org-Se species. I suggest removing this Figure. Also in addition to quoting average recoveries please provide the range for both extracts.

We think that the Figure 3 was misunderstood. Indeed, AEC-ICP-MS does not detect organic Se species. With this figure, we simply wanted to highlight the large proportions of unrecovered Se species using AEC-ICP-MS and the very good recovery for Se species with our new SEC-UV-ICP-MS/MS method. Given this comment and the general comment above, we replaced Figure 3 by a table (i.e., Table 1 in revised manuscript). In addition, we quoted now the ranges of Se species recoveries instead of providing the average recoveries: “*Considering all soils, the Se recovery obtained with SEC accounts for 58-109% of total Se in water extracts (Se_{water}) and 95-112% of total Se in NaOH extracts (Se_{NaOH}; Table 1)*” (L. 132-133 in revised manuscript).

Line 193: Figure b – should this be 5b?

Corrected (L. 203 in revised manuscript). Please note that Figure 5 is now Figure 4 in revised manuscript.

Line 201: ‘allowing estimation of’ not ‘allowing to estimate’

Corrected (L. 211 in revised manuscript).

Line 208: not only ‘are’ the association of As and Se to SOM different. Please amend the position of ‘are’

Corrected (L. 217 in revised manuscript).

Line 213: ‘associated with’ not ‘associated to’ (twice)

Corrected here and everywhere else in the manuscript and supplementary information (e.g., L. 202, 204, 218, 220, 222, and 226 in revised manuscript).

Line 215: please remove ‘the’ from before ‘so-called’

Corrected (L. 224 in revised manuscript).

Line 219: ‘has’ not ‘had’

Corrected (L. 228 in revised manuscript).

Line 225: ‘differs’ not ‘differ’

Corrected (L. 234 in revised manuscript).

Line 278: non-extracted? Not extracted? Unclear.

Here we meant soil Se that is not extracted by NaOH. To clarify, we rewrote the sentence as follows: “*In addition, the EF of soil Se that is not extracted by NaOH, which...*” (L. 285 in revised manuscript).

Line 290: ‘more’ or ‘greater’ not ‘stronger’?

We replaced “*stronger*” by “*greater*” (L. 297 in revised manuscript).

Line 310: ‘very close to’ not ‘very close from’

Corrected (L. 312 in revised manuscript).

Line 311: ‘Aside from’

Corrected (L. 313 in revised manuscript).

Line 346: collections

Corrected (L. 348 in revised manuscript).

Line 348: simplify to ‘can be combined with’ removing ‘directly hooked or’

Corrected (L. 350 in revised manuscript).

Line 462-476: Please clarify the isotope used as the post UV-detector spike. In Section 2.1.1 of the SI, Fig S4, Table S3 it is given as ^{77}Se whereas here the isotope discussed is ^{78}Se .

This information is correct. We used a ^{77}Se isotopically enriched standard as internal standard for the optimization of the SEC separation and then performed on-line isotope dilution using a ^{78}Se isotopically enriched standard when analyzing all Kohala soil extracts with the optimized SEC method. This is clarified in a newly added Table in the supplementary information (cf. Section 3.1 and Table S3 in revised supplementary information; p. 6-8).

Line 470: consisted 'of' not 'in'.

Corrected (L. 491 in revised manuscript).

Line 492: 'Aqueous standards' not 'liquid standards'

Corrected (L. 518 in revised manuscript).

Supplementary Information

Table S2: 'Fecrystalline' not 'Fecristalline' in column 11.

Corrected (See Table S2 in revised supplementary information, p. 5).

Section 2.1.1: Discussion of the choice of mobile phase is confusing. The justification for not using ammonium citrate is to reduce C content but you are adding MeOH (to enhance Se sensitivity I assume) which will increase C content. The discussion then continues to discuss ammonium phosphate as a mobile phase. Please clarify which eluents were tested and at what concentrations. Please also provide information on the extent of sensitivity loss during long runs.

We acknowledge that the discussion of the choice of mobile phase could be confusing. To improve the clarity of the technical aspects of our work (see also the answer to last comment of this review), we added a new table in the supplementary information (cf. Table S3 and section 3.1 in revised supplementary information, p. 6-8). This table details all SEC-UV-ICP-MS/MS parameters, including the tested eluents and concentrations, used during the SEC optimizations and during the analysis of all Kohala soil extracts. It is thus clear now that MeOH was added to the mobile phase only during the SEC optimization to evaluate the extent and impact of secondary hydrophobic interactions on the separation of Se and other elements. However, as described in supplementary section 3.2.3 (p. 12-13 in revised supplementary information), these tests demonstrated that hydrophobic interactions were minor and only resulted in slight shifts of retention times, meaning that the presence of MeOH in the mobile phase was not required for an optimal SEC separation of Se species in soil extracts. MeOH was therefore not added to the mobile phase of the optimal method used for analyzing all soil extracts in order to detect C by ICP-MS/MS. This is thus why we selected ammonium nitrate for the mobile phase over ammonium acetate, which would have impeded C detection by ICP-MS/MS.

Concerning the extent of sensitivity loss during long runs, we have added two plots in the supplementary information, which show the variations of the internal standards (^{45}Sc , $70 \mu\text{g L}^{-1}$ and ^{89}Y , $70 \mu\text{g L}^{-1}$) that were added post-column together with the ^{78}Se enriched isotope during the final analysis of all Kohala soil extracts. These plots can be found in Figure S11 together with an associated discussion (cf. section 3.5 "*Stability of the SEC-ICP-MS/MS analysis during long runs*" in revised supplementary information; p. 20). Briefly, the variations of ^{45}Sc and ^{89}Y during the analyses of all Kohala soil extracts (that lasted for up to 60 hours) were within $\pm 7\%$ for NaOH extracts and $\pm 5\%$ for water extracts, indicating that the loss in sensitivity during long SEC-UV-ICP-MS/MS runs was small. In any case, the use of on-line isotope dilution to quantify Se in SEC peaks corrected for the small change in sensitivity during the run.

Figure S3: Please clarify what the numbers given on the lines represent. Please check the r^2 value for polystyrene sulfonate in panel A – it appears to be an r^2 for a line through some of the data points? Also for Pullulan standards in Panel C.

We clarified these numbers as follows: "*On each plot, the number written aside each data point (above or below the trend line) represents the molecular weight, expressed in kDa, of each analyzed pullulan and PSS standard*" (cf. Figure S3 caption in revised supplementary information, p. 11). We kept this information because it is directly useful for readers who want to purchase the same standards that are named by the manufacturer with their molecular weight (i.e., readers won't have to convert the $\log(\text{molecular weight})$ from the y-axes of our figure into molecular weight). In addition, we corrected the r^2 and the trend lines of the pullulan standards eluted with the Shodex columns and of the polystyrene sulfonate (PSS) standards eluted with the Superdex peptide column (cf.

revised Figure S3, panels A and C). We also clarified that the trend line for the PSS standards with the Superdex peptide was calculated excluding the 30kDa polystyrene standard that elutes in the column void volume.

Section 2.1.5: Reference is made to phosphate extracts – please clarify. Only NaOH and water extracts are referred to elsewhere.

We thank the reviewer for pointing out this inconsistency. This reference to phosphate extracts is a “relict” from an earlier version of our manuscript. We did phosphate extractions for few of the studied Kohala soils and analyzed them by SEC-UV-ICP-MS/MS. However, because the obtained data did not bring more information than the water and NaOH extracts, we decided to remove these data from the manuscript. We thus removed the reference to phosphate extracts in the supplementary text.

Overall the technical aspects of the manuscript are quite challenging to follow because they are referred to in the results and discussion, separate materials and methods section which follows the discussion, and also in the SI where the testing and development of the method is described. To assist the reader I suggest inclusion of a table in the SI that details all the parameters tested to develop the method.

We agree that it is helpful to give all the details on tested SEC-UV-ICP-MS/MS parameters and their final selection in one table, which is now included in the supporting information (i.e., Table S3 in revised supplementary information, p. 7-8). To improve the presentation of the technical aspects of the manuscript further, we also revised:

- the section “*Se, S and As speciation in soil extracts by SEC-UV-ICP-MS/MS*” of the Materials and Method part in the main manuscript (L. 455-505 in revised manuscript).
- the section of the supplementary information providing an overview of the SEC optimization (cf. Section 3.1 in revised supplementary information, p. 6).

REVIEWERS' COMMENTS

Reviewer #2 (Remarks to the Author):

The authors have carefully addressed all my review questions, along with other reviewers' comments, in their revised manuscript. Indeed, the quality of this revised manuscript has been further improved. Thus, I endorse this manuscript for publication in Nature Communication.

Response to reviewer's comments (revision 2)

Reviewer #2 (Remarks to the Author):

The authors have carefully addressed all my review questions, along with other reviewers' comments, in their revised manuscript. Indeed, the quality of this revised manuscript has been further improved. Thus, I endorse this manuscript for publication in Nature Communication.

We would like to thank the reviewer for the positive feedback on our revised manuscript and again for their previous useful comments.